# The Role of Gravity Waves in the Mesosphere-Lower-Thermosphere Inversion Layers over Low-Latitude Using SABER Satellite Observations

**Chalachew Lingerew[1*], U. Jaya Prakash Raju[1]**

[1] Department of Physics, Washera Geospace, and Radar Science Laboratory, Bahir Dar University, Bahir Dar, Ethiopia

*Correspondence to*: Chalachew Lingerew (chalachewlingerew@gmail.com)

## Abstract

The Mesosphere and lower thermosphere (MLT) transitional region is a distinct and highly turbulent zone of the atmosphere. A transition mesosphere region is connected with dynamic processes, particularly gravity waves, as a causative of an inversion phenomenon. Understanding MIL (mesosphere inversion layer) phenomena is important under the influence of atmospheric waves for the understanding of middle and upper atmosphere dynamics for two primary reasons: stability and energy transfer. Mesospheric inversions have been the subject of numerous investigations, but their formation mechanisms are still poorly understood. In this article, an attempt has been made to investigate the upper and lower inversion phenomena and their causative mechanisms using long-term SABER observations in the height range of 60-100 km from 2005 to 2020 over a low-latitude region (3-15$^0$ N). The results indicate that the frequency of occurrence rate for the upper inversion is below 40%, whereas for the lower inversion, it is below 20%, indicating that the upper inversion is dominant over the lower inversion. The upper inversion exists in the height range of 78-91 km with an inversion amplitude of ~20-80 k and a thickness of ~3-12 km, whereas the lower inversion is confined in the height range of 70-80 km with an inversion amplitude of ~10-60 k and a thickness of ~4-10 km. Therein the gravity wave indicator potential energy depicts high energy (below 100 J/kg) in the upper mesosphere region (85 and 90 km) compared to the lower mesosphere region (70 and 75 km) with less than 50 J/kg. On account of Gws, the stability criteria from Brunt-Vaisala frequency ($N^2$) indicate instability in the upper mesosphere region with very low values relative to the lower mesosphere region. This result leads us to the conclusion that a high amount of gravity wave potential energy is a consequence of the high instability in the upper inversion relative to the lower inversion.

**Keywords.** MLT, Upper and Lower Inversions, Perturbed temperature, Causative gravity waves, Potential Energy, Brunt-Vaisala frequency, Instability.

## Introduction

The mesosphere dynamic regions act as a transition zone to the lower and upper atmospheric wave processes (tidal waves, planetary waves, and gravity waves). It is a well-known fact that atmospheric waves, especially gravity waves (GWs) generated from the lower atmosphere, propagate into the middle and upper atmospheres, break in the mesosphere region during propagation, and dissipate their energy and momentum into the background atmosphere, influencing the dynamics of the mesosphere thermal structure, global atmospheric circulation, variability, and even the MIL phenomenon (Lindzen, 1981; Holton, 1983). The mesospheric inversion layers (MILs) are a common feature that appeared to increase the mesosphere temperature variability. The MIL is a symptom (sign) of wave saturation in the mesosphere when the lapse rate is less than half of the dry adiabatic lapse rate (Sica et al., 2007). Temperature inversions have been omnipresent features in the mesosphere regions for decades, and they have been comprehensively studied in the past by using all sorts of available techniques (e.g., lidar, radar, rocket sonde, and satellite) over different geographic locations. (Sivakandan et al., 2014) use the TIMED/SABER kinetic temperature data to study the occurrence of mesospheric inversions and their characteristics over equatorial Indian region (0 to 10∘ N and 70 to 90∘ E) for the year 2002 and 2008, which is not considering the causatives. In the present work, we investigated the causatives, atmospheric waves particularly gravity waves on an inversion.

The gravity waves (GWs) momentum and energy deposition is thought to be the principal mechanism driving large-scale circulation and coupling of distinct atmospheric layers, as well as inversion phenomena (Fritts and Alexander, 2003; Lindzen, 1981; Smith, 2012). Further, the gravity wave-breaking influence on mesosphere dynamics is an attempt to demonstrate their impacts on the inversion phenomenon over mid and high latitudes (Gan et al., 2012; Walterscheid and Hickey, 2009; Collins et al., 2011; Szewczyk et al., 2013). Observational and modeling approaches have been used to investigate GWs as the causative of inversions (Fritts, 2018; Collins et al., 2014; Sridharan et al., 2008; Ramesh and Sridharan, 2012; Ramesh et al., 2013, 2014, 2017). The effect of gravity waves in the mesosphere inversion based on temperature variability is studied particularly over the mid- and high-latitudes (Singh and Pallamraju, 2018; Fritts et al., 2018) but not yet sufficiently understood. As a result, the inversion phenomenon and their causative investigation is the topic of numerous studies in the mesosphere dynamics.

Regarding the low-latitudes, there are very less number of studies on the temporal (time) and spatial (altitudinal, latitudinal, and longitudinal) variability of the mesosphere inversion phenomenon associated with atmospheric waves particularly gravity wave activity. This motivates us to investigate the mesosphere inversion phenomenon and its association with gravity wave activity, along with stability criteria using Brunt-Vaisala frequency ($N^2$) over the low latitudinal band (3-15$^0$ N) using long-term SABER observations during 2005-2020. This is organized as follows: The data and method of extracting the mesosphere inversion phenomenon are presented in Section 2, and their results are described in Section 3. Finally, Section 4 presents the conclusions.

## 2.    Observation and Data analysis

## 2.1    SABER Observation

The TIMED/SABER satellite was launched on December 7, 2001, to set on an elliptical orbit at an altitude of about 625 km with an inclination of 74$^0$ from the equator. The SABER instrument makes 15 orbits; each orbit takes 97 minutes (1.6 h) and provides about 1400 profiles per day; each profile takes 58 seconds. This TIMED/SABER satellite provides temperature profiles with good spatial and temporal resolution to investigate mesosphere dynamics and their atmospheric wave processes. SABER temperature data has been widely used to investigate the typical thermal structure and dominant dynamical processes in the mesospheric region (Garcia et al., 208, Gan et al., 2012, 2014; Bizuneh et al., 2022; Lingerew et al., 2023). For vertical temperature measurement, SABER provides an accuracy of 1 to 2 K between 15 and 60 km, decreasing to 5 K below 85 km, while the error increases with altitude from 6.7 K to 10 K near 100 km (Rezac et al., 2015). The valuable nature of SABER observations for the study of the middle atmosphere is well documented in previous research (Meriwether and Gerrard, 2004; Fechine et al., 2008; Dou et al., 2009; Gan et al., 2012; France et al., 2015). In the present, we have used the latest version of SABER temperature data over low-latitudes. The SABER vertical temperature profiles were taken in the range of 60-100 km altitude during the period January 2005-December 2020 over (3-15$^0$N) latitude and (33-48$^0$E) longitude regions. The monthly mean SABER temperature data of the mesosphere and lower thermosphere (MLT) region is presented as shown in Figure 1. The monthly mean temperature of the MLT (60-100 km) region shows a maximum temperature of 200-240 K in the height range of 60-70 km, with the minimum temperature declining to around 160-180 K in the height range of about 95-100 km throughout all over the period.

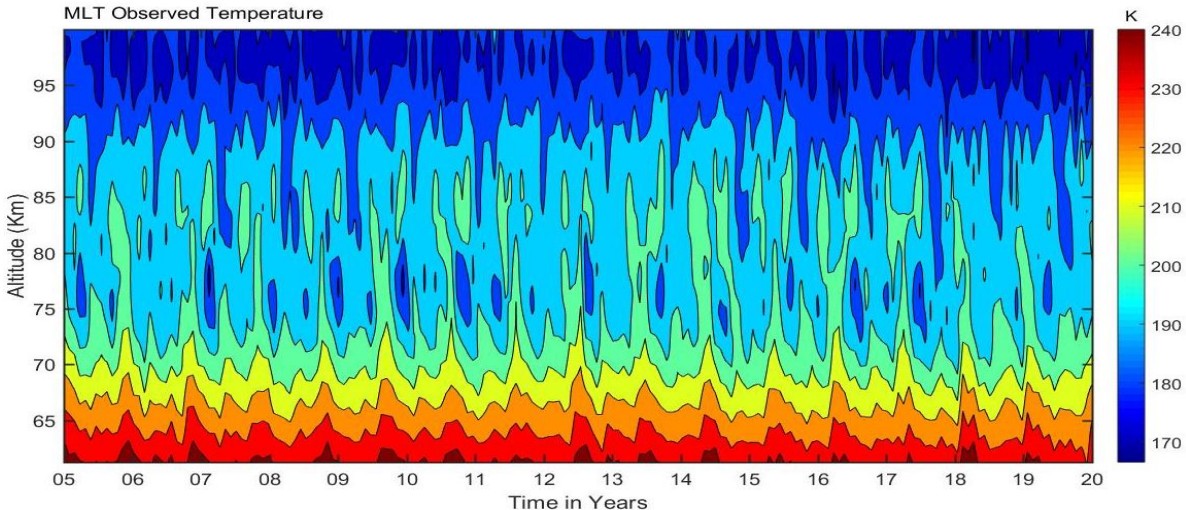

92

**Figure** 1. The monthly mean MLT temperature variability in the height range of 60-100 km during December 2005-January 2020 over the low-latitude.

## 2.2    Analysis Technique

The mean thermal structure of the Earth's middle atmosphere is characterized by a negative temperature gradient. However, there are several reports showing positive temperature gradients in the mesosphere which are in contrast to the ideal situation of negative gradients (Meriwether and Gardner, 2000; Gan et al., 2012). This kind of phenomenon is known as "mesospheric inversion layer (MIL)". MILs were identified based on following procedure outlined by Leblanc and Hauchecorne (1997) and Fechine et al. (2008) and which is briefly presented here. This procedure has been applied in many previous studies investigating the phenomenon of mesospheric inversion (Leblanc et al., 1998; Meriwether and Gardner, 2000; Duck et al., 2001; Duck and Greene, 2004; Cutler et al., 2001; Siva Kumar et al., 2001; Ratnam et al., 2003; Gan et al., 2012). Inversions of this MLT temperature are identified based on their characteristics thickness (the altitude difference between the point of warm & cool), while the temperature difference between the point of cooling and warming is termed as amplitude of the MIL (Meriwether and Gardner, 2000) as shown in figure 2, which have a positive value between the top and bottom levels.

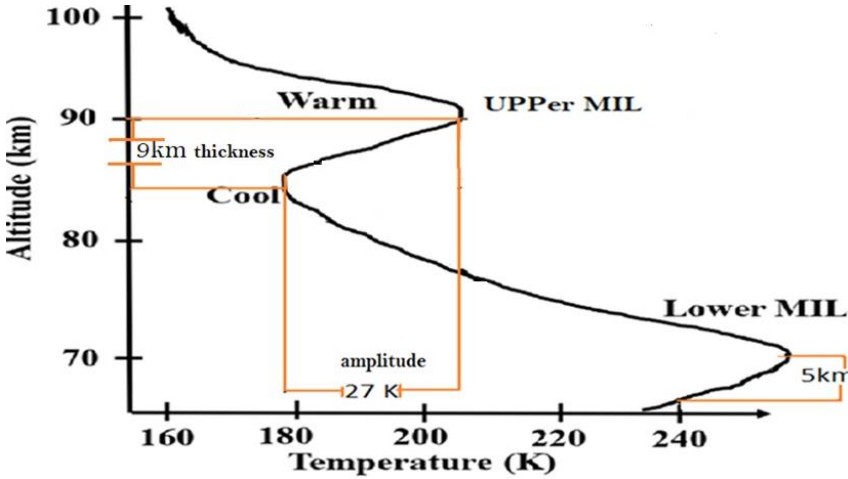

Figure 2. Schematic of upper and lower mesospheric inversion layers shown in the temperature profile for the MLT regions (Adapted from Meriwether and Gerrard, 2004).

In this regard, the upper and lower mesosphere inversions were identified in the procedure of Leblanc and Hauchecorne (1997) based on the characteristics of the temperature inversions using the following criteria: (1) The bottom level of the lower and upper inversions is above 70 and 80 km, and its top level of inversion is below 80 and 92 km, respectively; (2) the amplitude is considered larger than 5 K; and (3) the thickness is greater than or equal to 2 km following the procedure. Based on this sequence of temperature inversion, the diagnostic technique is applied to the SABER observed data during the period 2005-2020 over low latitudes to investigate the causative influence of atmospheric gravity waves (Gws). Inversions that satisfy the above-mentioned criteria are considered significant. As well as identifying inversions, the frequency occurrence rate (%) of mesospheric inversion layers (MILs) is derived during the period 2005–2020 in the upper and lower MLT regions. The occurrence rate of the frequency (percentage) is estimated based on dividing the monthly inversion days for each month (dates of a month) of 16-year (2005–2020) observation data.

The inversion of the mesosphere temperatures is related to the instabilities. Hence, we are going to derive the hourly atmospheric gravity waves via the Brunt-Vaisala frequency ($N^2$). Whereas another important concept to estimate the Brunt-Vaisala frequency is the potential temperature ($\theta$), which stands for the air parcel's temperature when it is displaced adiabatically to a standard pressure level, $p_0$, from the current pressure level, p, the first law of thermodynamics:

$$\frac{dT}{T} = \frac{R}{c_p}\frac{dp}{p} \Rightarrow \int_T^0 \frac{dT}{T} = \int_p^{p_0} \frac{R}{c_p}\frac{dp}{p} \qquad (1) \text{ it yields}$$

$$\theta = T\left(\frac{p_0}{p}\right)^{R/c_p} \qquad (2)$$

Therefore, the motion of the vertical atmospheric air parcel can be described by (Liu, 2011; Vadas
and Fritts, 2005) as follows in equation (2.3) to calculate the Brunt-Vaisala frequency of the parcel
due to the Buoyant and gravitational forces acting on the parcel:

$$\frac{d^2s}{dt^2} = -g\,\frac{\rho - \rho_0}{\rho}\,\sin a \qquad (3)$$

Based on the hydrostatic equation, $\rho = \rho_0$, and $p = p_0 \Rightarrow \frac{\partial p}{\partial z} = \frac{\partial p_0}{\partial z} = -g\rho_0$  (4) and the ideal gas
law, $\rho = p/RT = p_0/RT$ gives the parcels motion of an equation:

$$\frac{d^2s}{dt^2} = -\frac{g}{\rho}\left(\frac{d\rho}{dp}\frac{\partial p_0}{\partial z} - \frac{\partial \rho_0}{\partial z}\right)z \qquad (5)$$

Following the same approach using the hydrostatic equation (4) and adiabatic equation (6)

$$dln\rho = \frac{dlnp}{\gamma}\ ,\gamma = c_p/c_v \qquad (6)\ \text{yields}$$


$$\frac{d^2s}{dt^2} = -\frac{g}{\rho}\left(\frac{\rho}{\gamma p_0}\frac{\partial p_0}{\partial z} - \frac{\partial \rho_0}{\partial z}\right)z = g\left(\frac{\partial ln\rho_0}{\partial z} - \frac{1}{\gamma}\frac{\partial lnp_0}{\partial z}\right)z \qquad (7)$$

For the ideal gas law of $p = \rho RT$, the natural logarithm is taken for altitude, z on both sides, yielding

$$\frac{\partial ln\rho}{\partial z} = \frac{\partial lnp}{\partial z} - \frac{\partial lnT}{\partial z} \qquad (8)$$

Then after, the potential temperature ($\theta$) of the parcel is calculated as follows based on the equation

147   (2):

$$\frac{\partial ln\theta}{\partial z} = \frac{\partial lnT}{\partial z} - \frac{R}{c_p}\frac{\partial lnp}{\partial z} = \frac{1}{T}\left(\frac{\partial T}{\partial z} + \frac{g}{c_p}\right) = \left(1 - \frac{R}{c_p}\right)\frac{\partial lnp}{\partial z} - \frac{\partial ln\rho}{\partial z} \qquad (9)\ \text{to derive the Parcels}$$

acceleration based on equations (7) to become:

$$\frac{d^2s}{dt^2} = -g\,\frac{\partial ln\theta_0}{\partial z}z\,\sin a = -g\,\frac{\partial ln\theta_0}{\partial z}ds.\sin^2 a \qquad (10)$$

Whereas by introducing the frequency, N, with $N^2 = g\,\frac{\partial ln\theta_0}{\partial z}$
The Brunt-Vaisala frequency, $N^2$ is calculated based on the following mathematical formulation
used to characterize atmospheric stability.

$$N^2(z) = \frac{g(z)}{T_0(z)}\left(\frac{\partial T_0(z)}{\partial z} + \Gamma_d\right) \qquad (11)$$

Where g is the acceleration due to gravity, N is the Vaisala frequency, $T_0$ is the background
temperature, estimated based on the third-order polynomial fitting, $\Gamma_d = \frac{g}{c_p}$ is the adiabatic lapse
rate, and $c_p = 1004\,J\,K^{-1}\,kg^{-1}$ is the specific heat capacity of the atmosphere at constant
pressure. When Vaisala frequency $N^2$, is statically positive, the atmosphere is stable. While the
frequency $N^2$, is negative, the atmosphere is unstable, in which the atmospheric lapse rate, $\Gamma =$
$-\frac{\partial T}{\partial z}$ is larger than the adiabatic lapse rate, $g/c_p \approx 9.5 \; K \; km^{-1}$, the atmosphere is unstable.
In the meantime of estimating the Brunt-Vaisala frequency, the third-order polynomial fit of the
least squares has been applied to the SABER observed temperature (T) profile to estimate the
background temperature ($T_0$) following the procedure Leblanc and Hauchecorne (1997).
Succeeding the estimations of the perturbed temperature ($T_p$) from equation (2), were identified,
which is estimated by subtracting the background from the observed temperature data (T).
$$T_p = T - T_0 \qquad\qquad (12)$$
After estimating the perturbed temperature ($T_p$), a one-hour cut-off frequency of the low-pass band
filter is applied on the perturbed temperature to calculate the atmospheric gravity wave potential
energy ($E_P$) by removing the planetary and tidal wave impacts or contribution in the perturbed
temperature (John and Kumar, 2012).
$$E_p(z) = \frac{1}{2} \left( \frac{g(z)}{N(z)} \right)^2 \left( \frac{T_p(z)}{T_0(z)} \right)^2 \qquad\qquad (13)$$
The potential energy of the waves is a function of altitude, z, which is utilized to determine the
impact of atmospheric gravity waves on atmospheric dynamics.

## 3. Results and discussion

### 3.1 Identification and Characteristics of the Lower and Upper MLT Inversion

The daily SABER observed temperature profiles of the upper and lower mesospheres from 2005
to 2020 over low latitudes are depicted in the form of contours in Figure 3(a and c) in the range
between ~ (180-220 K). The lower panel of Figure 3(b and d) shows daily inversion temperature
profiles in the range of 180–225 K, indicating temperature is maximum in the inversion day
observed temperature at lower and upper regions when compared without considering the
inversion day observed temperature in Figure 3(a and c).

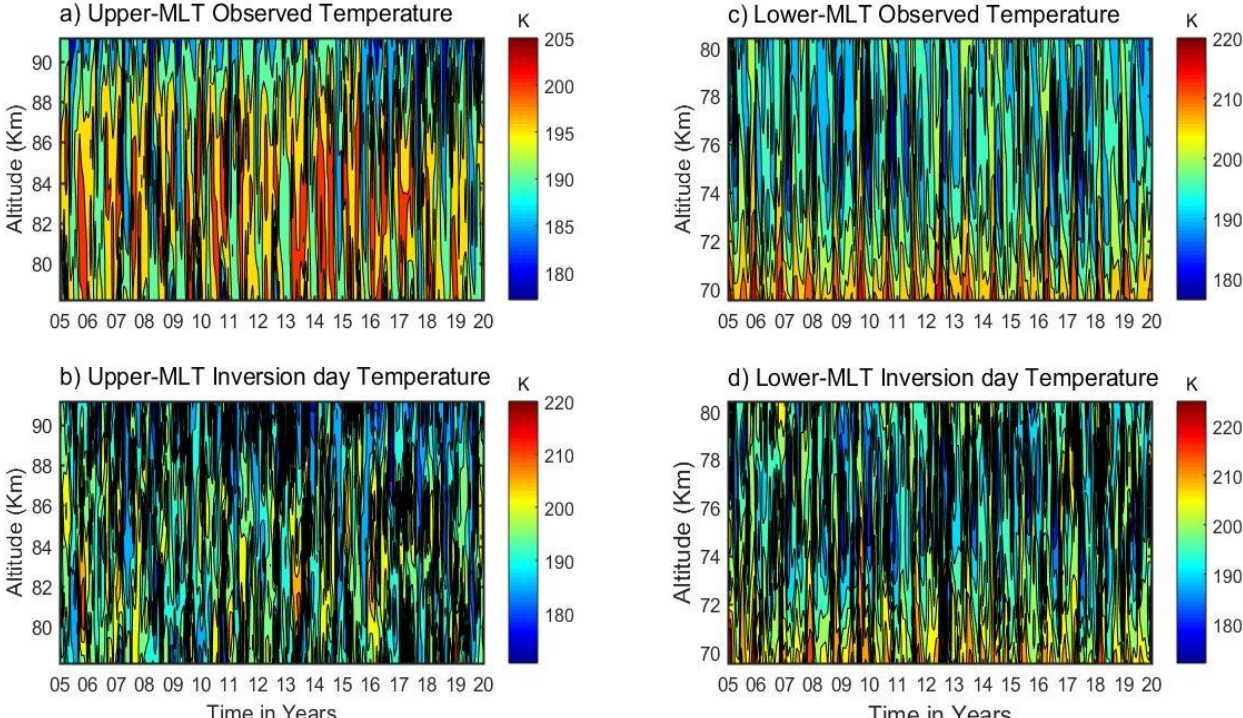

**Figure** 3. The upper and lower mesosphere observed temperatures in the first horizontal panel at (a and c) with their inversions in the second horizontal panel at (b and d).

The upper panel on the left side of Figure 3(a) represents the upper mesosphere observed temperature, which is depicted in the range ~(180-205 K) at the height around ~80-90 Km, and the right upper panel of Figure 3(c) represents the lower mesosphere observed temperature in the range around ~(180-220 K) at the height around ~70-80 Km. Whereas, Figure 3(b) depicts an upper-temperature inversion about ~(180-220 K) at an altitude of ~(80-90 Km), while Figure 3(d) shows a lower-temperature inversion about ~(180-225 K) at a height of ~(70-80 Km), indicating a temperature gradient is occurred from negative to positive due to external or internal drivers, which might be atmospheric gravity waves, chemical reactions or solar radiations. The first observation of MIL was carried out by a rocket-falling experiment, which shows temperature inversion layers have been normally detected with maximum values in the mesosphere and lower thermosphere (Schmidlin, 1976). Our findings of the lower inversions in the range of (70-80 km) tend to approach the reports by Sivakumar et al. (2001), which show that the base of the lower mesospheric inversion layer (MILs) lies in the range of (73-79 km), as well as the Gan et al. (2012) seasonal variations of MIL in the planetary waves as a caustive over low-latitudes using the SABER observations. Whereas Sivakandan et al., (2014) also investigated the lower and upper

mesospheric inversions in the altitudinal ranges from 60-105 km over low latitudinal regions,
which nearly coincides with our work results.

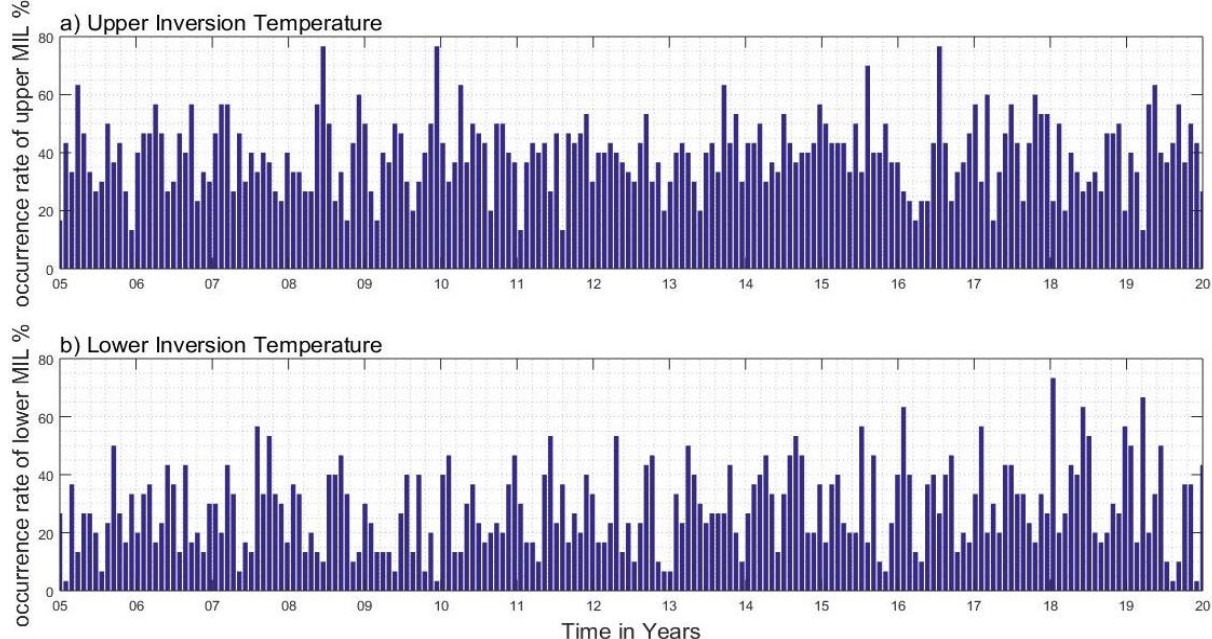

**Figure** 4. The frequency occurrence rate (percentage) of the (a) upper and (b) lower inversion
temperatures during 2005-2020 over low latitudes.
The frequency occurrence rate (%) of mesospheric inversion layers (MILs) were investigated as
displayed in the form of a histogram in Figure 4(a) for the upper MIL and in Figure 4(b) for the
lower MIL. The mean frequency occurrence rate of the upper inversion is approximately below
40%, whereas their maximum occurrence rate of inversion lies between 60% and 78%, particularly
in the years 2008, 2010, and mid-2016. While the mean frequency occurrence rate of the lower
inversion is below 20%. As a whole, the occurrence rate of the upper inversion is relatively high
compared with the lower inversion, which could be related to atmospheric wave activities,
particularly gravity wave activity. In this regard, Hauchecorne et al. (1987) and France et al. (2015)
show the impacts of Gws on the upper and lower mesosphere inversion variability. Not only this,
Gan et al. (2012) also found the seasonal variation of MILs over the low latitudes under the
causative planetary waves. As those previous scientific results of the occurrence rate the inversions
of the lower and upper MLT regions were investigated based on their characteristics amplitude
and thickness in Figrue 5.
Before examining the effects of Gws on the MLT regions of an inversion, Figure 5 depicts the
inversions of mesosphere temperature variability in terms of base height, amplitude, and thickness.
The frequency occurrence of amplitude, thickness, and base height of inversion variability in the
form of the histogram along with the best-fit red lines of the Gaussian distribution are presented
in Figure 5. The observed distributions coincide with Gaussian curves (best fits), indicating that
the number of MILs is distributed over their attributes according to normal laws, implying that the
representations are real-valued random variables. Figure 5 of the left vertical column, three rows
represent a histogram of (a) amplitude, (b) thickness, and (c) the base of the upper MIL
phenomenon, along with their statistical metrics mean and standard deviations (SD). Whereas the
corresponding three rows of the right vertical column represent (d) amplitude, (e) thickness, and
(f) the base of the lower MIL phenomenon.

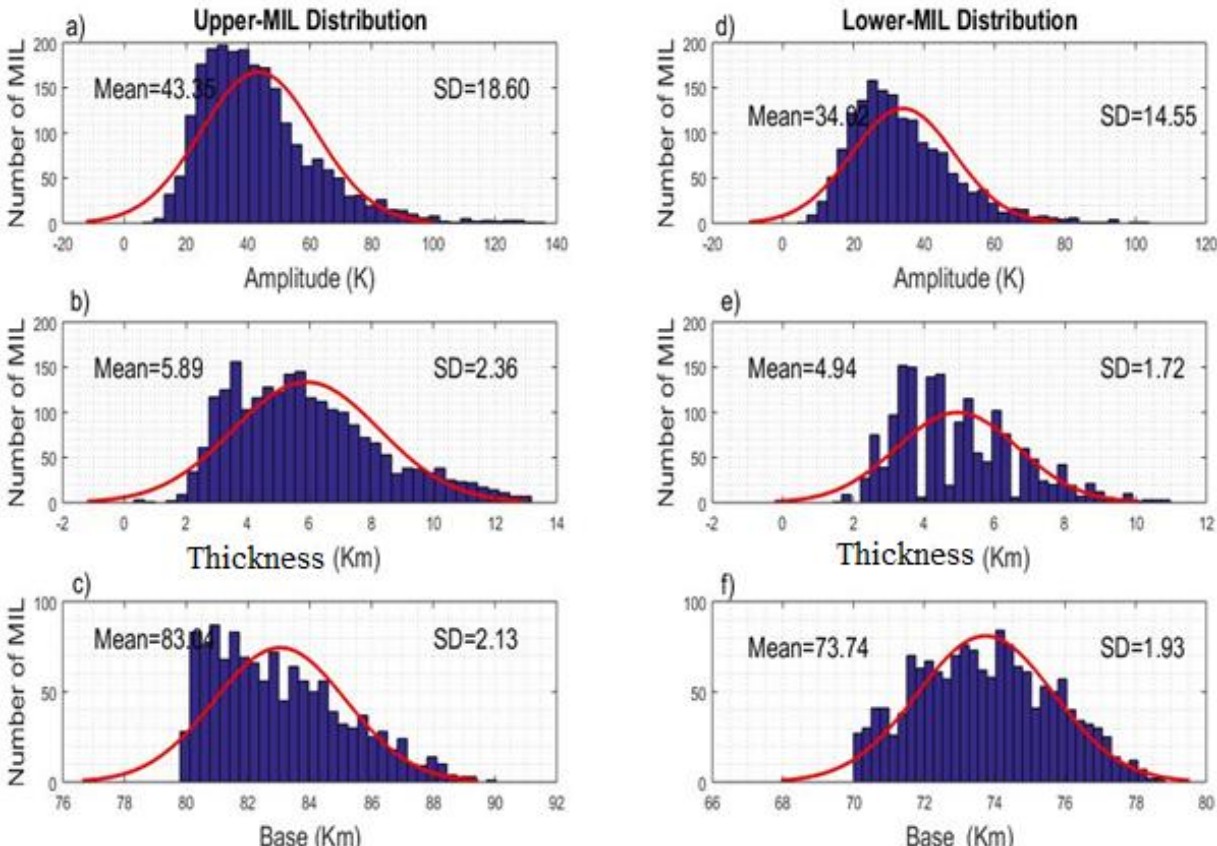


**Figure** 5. Histogram occurrence of mesosphere inversions. The first vertical panel represents the
upper inversion distribution of (a) amplitude, (b) thickness, and (c) base, and the corresponding
distribution in the second vertical panel is the lower inversion of (d) amplitude, (e) thickness, and
(f) base over the low latitude during the period 2005–2020.
The amplitude of upper inversion variability in the left vertical panel in Figure 5(a) exists in the
range between 20 and 80 K, with a peak value of 38 K following a Gaussian distribution with large

standard deviations (SD), 18.6. The thickness of the inversion layer for upper MILs has existed in the range of 3-9 K, with the most probable value of 5.5 K and a low standard deviation 2.3 of (Figure 5(b)). The base height of the upper MIL in Figure 5(c) ranges from ~80 to 90 km, with a peak value of a large number of upper mesospheric inversions occurring at a base height of around 83 km in a lower standard deviation (SD) 2.13. The number of upper inversions all over the period 2005–2020 at a height of 82 km is the highest relative to the rest in the range between 80 and 90 km. Such maximum mean to fit of Gaussian distribution may be the reason for the gravity wave breaking is that it dissipates energy as a causative factor for an inversion, while the wave generated from the lower to the upper atmospheric region as well as the impacts of the solar flux generated from the upper solar system. Whereas, the lower inversion amplitude is depicted in the range between 10 and 60 K with a peak of 25 K and standard deviations (SD) 14.5 of in Figure 5(b) in the right vertical panel. The thickness of an inversion has appeared in the range of 3-8 Km, with the most probable value of 3.8 Km and a low standard deviation (SD) 1.72 of (Figure 5(d)). The base height of the lower inversion of Figure 5(f) is in the range of 70 and 80 km, with a peak value of around 74 km, showing a lower standard deviation (SD) of 1.93. In the earlier investigation, from the Indian sector, Sivakandan et al. (2014) reported amplitudes in the range (14–39 K) during 2002 and (15–42 K) in 2008, whereas their thickness was in the range of (2.7–7.5) during 2002 and (2.8–7.3) in 2008 to characterize the mesospheric inversion variability under the influence of solar flux, which agrees well with the present investigation. This comparison reveals that there is no significant variation in characterizing the mesosphere inversion based on amplitude and thickness over the low-latitude region in the altitude range of 60 to 90 km.

## 3.2    Latitudinal Variations of Mesospheric Inversion Layers (MILs)

In this section, the spatiotemporal (latitudinal-time) variability of the upper and lower mesosphere inversion phenomena is characterized in the contour plots of time **vs.** latitude in Figures (6 and 7) respectively, based on amplitude, thickness, and base height over the low-latitude band ($3\text{-}15^0$) during 2005–2020. The Upper MILs phenomenon is observed around 80–90 km, with the maximum amplitude in the range of 90–120 K over the latitude bands ($5\text{-}12^0$ N) during 2005, 2007, mid-2011, 2013, 2015, 2016, mid-2019, and 2020 (Figure 6(a)). The inversion thickness depicted in the second horizontal panel, as shown in Figure 6(b), is displayed with a maximum range of ~($8\text{–}12$ Km) over the entire latitudinal region ($3\text{-}15^0$ N). Figure 6(c) displays the relative maximum

inversion base height around ~(84-88 Km) in the latitudinal range between 4 and $14^0$ N during
2006, 2008, 2010, 2012, 2016, and 2018.

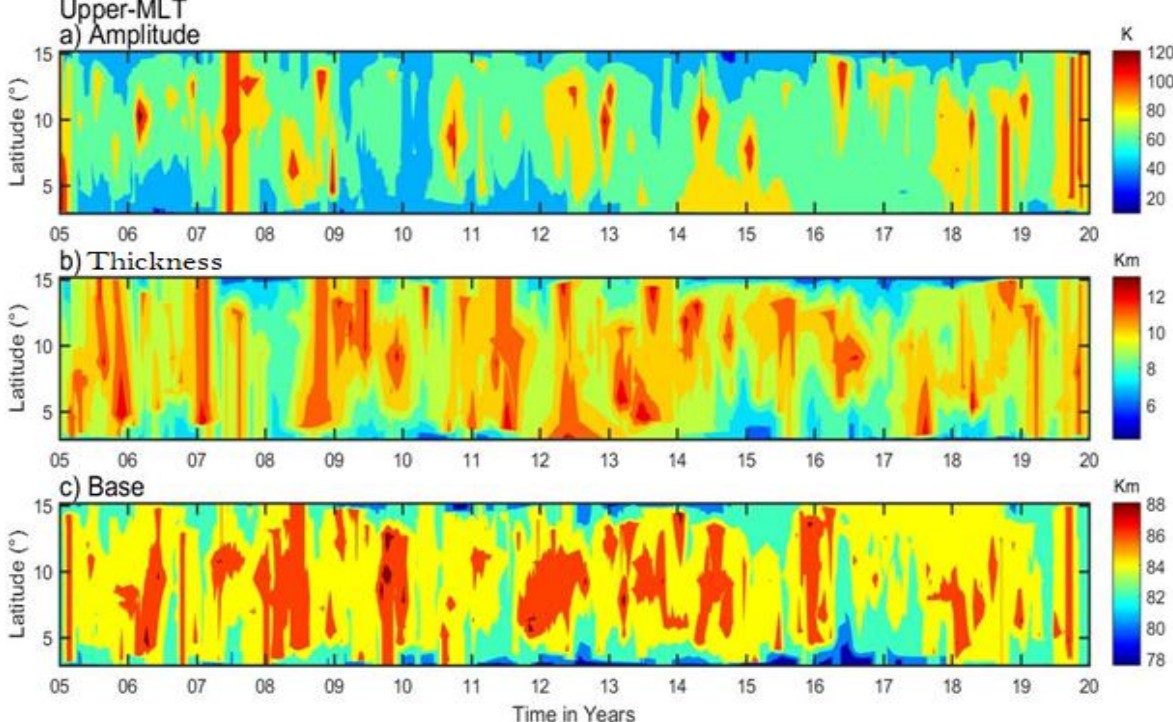

**Figure** 6. The daily upper inversions (~80-90 km) of (a) amplitude, (b) thickness, and (c) base
height during 2005-2020 over latitudinal variation.
Similarly, the latitudinal variations of the lower inversion (MILs) phenomenon based on their
characteristics amplitude, thicknesses, and base height are depicted in the form of contour plots of
time **vs** latitude in Figure 7(a, b, and c), respectively, over an altitudinal range around ~(70-80 km).
The lower inversion amplitude is depicted in the range of ~30-60 k over all latitudinal bands except
the maximum range of ~(80-100 k) during 2013, 2015, 2016, and 2019 in different latitudinal
regions enclosed in the range between 5 and $14^0$ N. Figure 7(b) displays the inversion thickness of
5-7 km over the entire latitude band, except for the maximum thickness of 8-10 km. The inversion
of base height (76-80) is depicted in Figure 7(c) over all latitudes and periods except 2008, 2014,
and mid-year 2018 with maximum base height. Figures 6 and 7, clearly show that the high
amplitude and thickness of the upper inversion in comparison with the lower inversion indicate a
highly dynamic phenomenon over the upper mesosphere region.

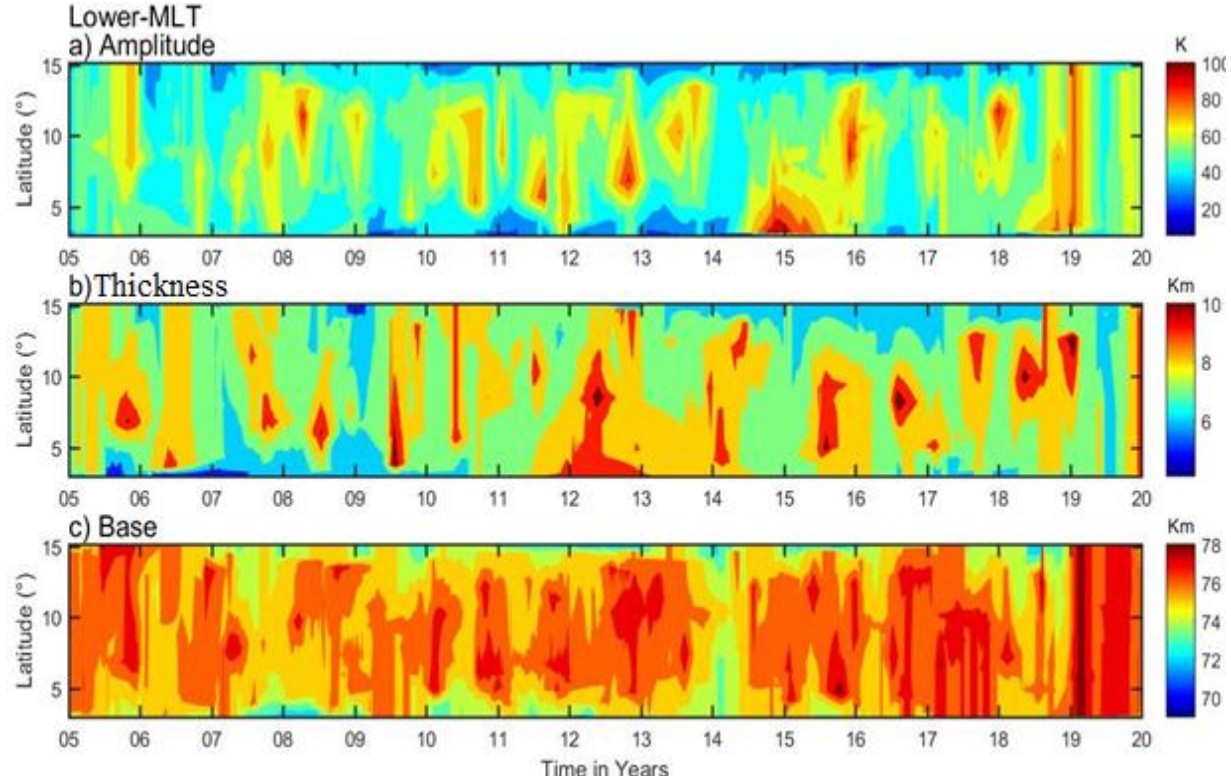

**Figure** 7. Same as Figure 5, but for the lower mesosphere inversions (~70- 80 km).
From Figures 6 and 7, it is observed that the upper inversion amplitude and thicknesses show high
values in comparison with the lower inversion, indicating a highly dynamic phenomenon over the
upper mesosphere region. Satellite measurements have a significant contribution to the
information on latitudinal variations in MILs. The global climatology of MILs observed by
TIMED/SABER shows that MILs also occur at low latitudes (Gan et al., 2012).
**3.3    Separations of the Perturbed Temperature in the Mesosphere Region**
The perturbed temperature profiles ($T_P$) in the upper and lower mesosphere inversions during the
period of 2005-2020 can further be used to calculate their factors' potential energy of gravity waves
and the Brunt-Vaisala frequencies ($N^2$). The procedure for calculating perturbation temperature
($T_P$) is mentioned in the methodology part.
First, the upper-temperature inversion profiles are identified in the MLT region during the entire
observational period of 2005-2020, as displayed in the contour plot of Figure 8(a). It is noted that
the observed temperature is in the range of ~170-220 K with less detectable variability. Based on
this inversion temperature profile, the background temperature ($T_0$) is calculated by applying a 3rd-
order polynomial fit as shown in the corresponding contour plot of Figure 8(b). This background
temperature displays identifiable periodic variability in the range of ~195-197 K around ~82-87
km. While the perturbed temperature profiles ($T_p$) are based on the difference between the
observed inversion temperature (T) and the corresponding background temperature profiles (To),
they display in the range of -25 to +25 K, as shown in Figure 8(c).

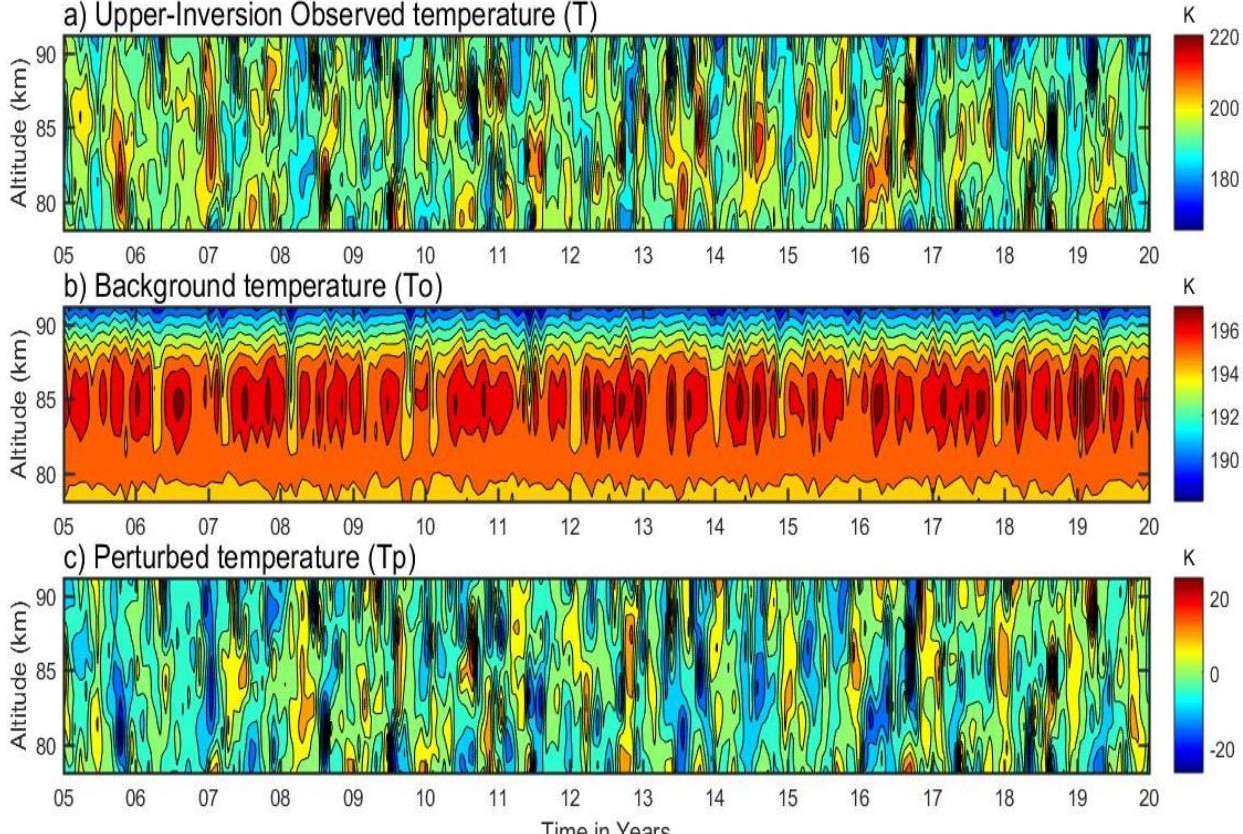

**Figure** 8. The upper mesosphere temperatures in the vertical panel are: (a) inversion day observed
temperature; (b) background temperature; and (c) perturbed temperature in the upper mesosphere
region.
A similar procedure has been applied to calculate the perturbed temperature ($T_p$) as well as the
observed and background temperature from 2005 to 2020 in the lower mesosphere region, and
their corresponding contours are displayed in Figure 9(a-c). The observed temperature of lower
inversion in Figure 9(a) depicted a range of ~170-220 K and the background temperature of lower
inversion in the range of ~ 195-210 K with their maximum values of ~200-210 K over the height
of ~70-72 Km as shown in Figure 9(b). Whereas the perturbed temperature in Figure 9(c) is
presented in the range between -25 and 20 K. It is noted that the upper mesosphere perturbed
temperature is at its maximum compared to the lower mesosphere region, which may be due to a
high dynamic phenomenon.

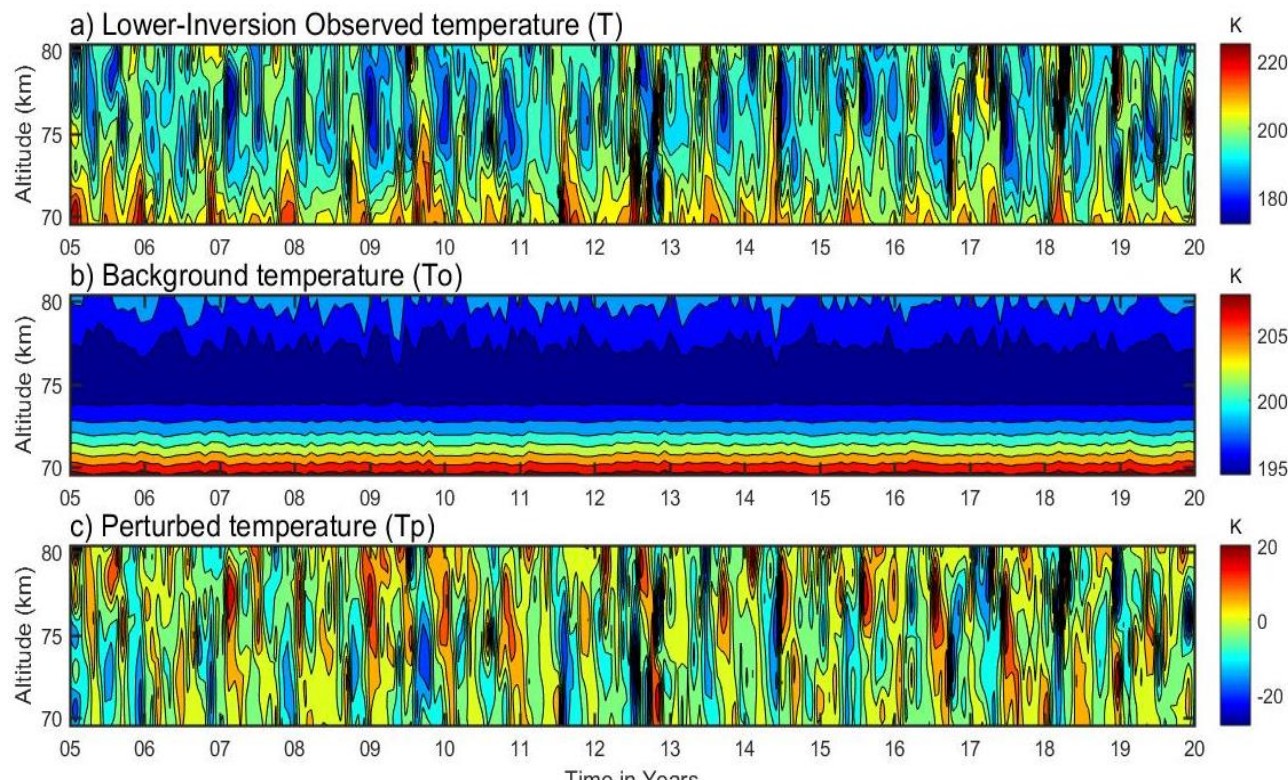

a) Lower-Inversion Observed temperature (T)

b) Background temperature (To)

c) Perturbed temperature (Tp)

Time in Years

**Figure 9**. Same as Figure 7, but for the lower mesosphere atmospheric region.

### 3.4    Effects of Gravity Waves on Mesosphere Inversions and Associated Instability

Atmospheric waves (gravity waves, planetary waves, and tidal waves) exist in different layers of the atmosphere and are generated by different mechanisms. Gravity waves are of local or regional dimensions, whereas the other two waves are of global extent. This dynamical gravity wave motion is a restoring force of gravity acting downward and buoyancy acting upward on vertically displaced air parcels from the troposphere/stratosphere through the upper thermosphere. These propagated gravity waves can be distributed from their source regions across the atmosphere and become saturated at the critical upper atmospheric level, particularly over the low latitudes. Thereby, the vertically propagated waves were breaking and dissipating to transfer their energy and momentum into the atmospheric background field, thus considerably affecting the structure and variability of the atmosphere, as shown in Figure 11, as well as the results of (Holton et al., 2003; Holton and Hakim, 2013) waves potential energy affecting the atmospheric temperature inversions. The  gravity wave propagation at the saturation stage  is broken in the upper region to dissipate their energy, which impacts the normal mesospheric temperature by increasing its temperature with elevation, known as an inversion. This is the reason the gravity wave potential energy is connected with an inversion.

In this section, an attempt has been made to investigate the longitudinal variability of gravity waves' contribution to the mesospheric inversions (MILs) phenomenon by calculating potential energy and their instability based on Bruent-Vaisala frequency ($N^2$) using perturbed temperatures. Before deriving the waves' potential energy from the perturbed temperature ($T_p$), a one-hour interval cut-off-frequency of low-pass band filter is applied on a perturbed temperature ($T_p$) during the period 2005-2020 at selected heights of 90, 85, 75, and 70 km, as depicted in Figure 10 (a, b, c, and d). The reason behind using the low-pass band filter is to eliminate/remove the unwanted influence of long-period oscillations on an inversion such as tidal or planetary waves from the gravity wave (Gw). The effects of the low-pass band filter are visible in Figure 10(a and b) for the upper mesosphere region at 90 and 85 km and in Figure 10(c and d) for the lower mesosphere region at 75 and 70 km. The amplitude of the perturbed temperature is reduced to the range around ~(-10 to 10 K), and the data is smoothed by eliminating higher frequencies.

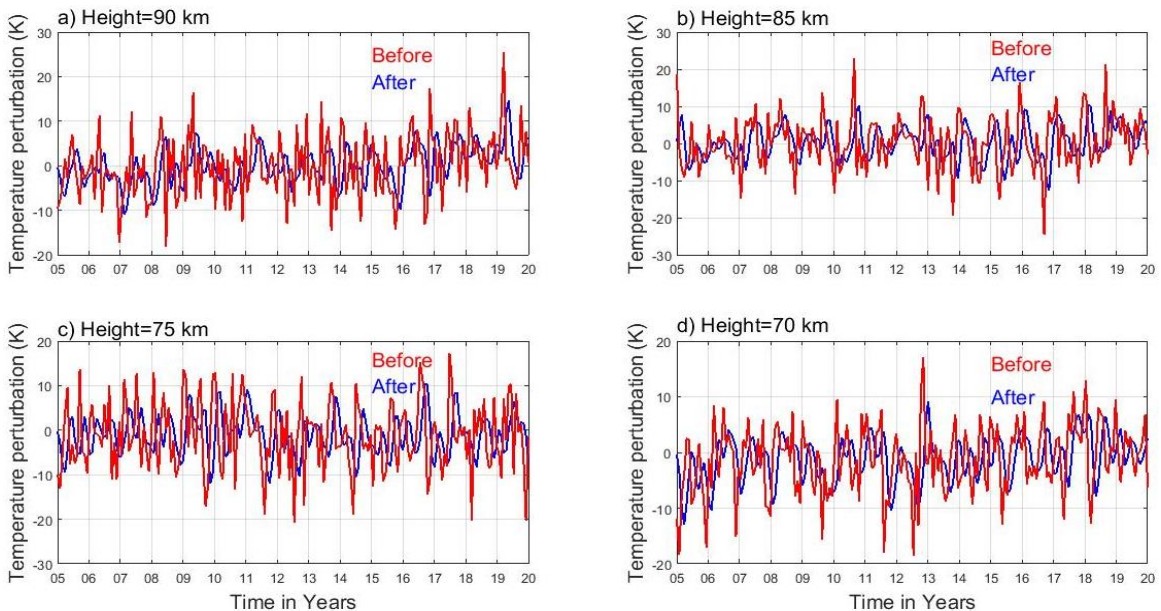

**Figure** 10. Perturbed temperature profiles before (red color) and after (blue color) applying the low-pass band filter for the upper (85 and 90 km) and lower (70 and 75 km) regions.

By using the time series of filtered perturbed temperature data at selected heights of 90, 85, 75, and 70 km, the potential energy ($E_p$) is constructed based on the formula mentioned in the methodology section, since the gravity wave activity is projected by the potential energy calculation as described from numerous authors (Tsuda et al., 2000; Wang and Geller, 2003; Liu et al., 2014; Thurairajah et al., 2014). The spatiotemporal variability of gravity wave potential

energy is shown in Figure 11(a and b) for the upper mesosphere region at (90 and 85 km) and
Figure 11(c and d) for the lower mesosphere region at (75 and 70 km).

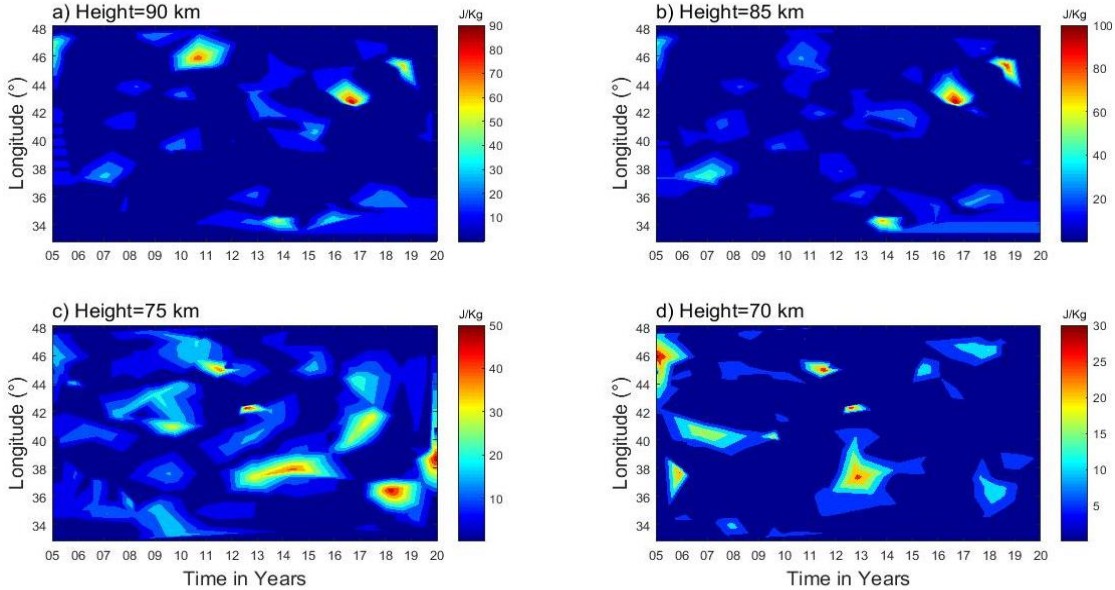

**Figure** 11. Gravity wave potential energy for the upper (85 and 90 km) and lower (70 and 75
km) mesosphere regions.
In this investigation, the maximum gravity wave potential energies were observed in the range of
around ~70–90 J/kg over the longitudinal regions of 45-47$^0$ E, 43$^0$ E, and 44$^0$ E during 2011, 2017,
and 2019 (Figure 11(a)) for upper mesosphere inversions at 90 km, whereas the potential energy
of gravity waves around ~10–60 J/kg is presented all over the longitudinal region from 33-48$^0$
E. While the maximum potential energy ~(70-100 J/kg) is observed as shown in Figure 11(b) over
the longitudinal (340, 440, and 460) regions during 2014, 2016, and 2018 at 85 km. The minimum
potential energy of gravity wave between 20 and 70 J/kg appears over the longitude (33-48)
regions. However, Figures 11 (c and d) show the lower MLT regions of gravity wave potential
energy at 75 and 70 km, respectively. At a height of 75 km allocated in Figure 11(c) is a relative
maximum potential energy appeared in the range of 40-50 J/kg over the longitudinal (46$^0$, 42$^0$, 40$^0$,
37$^0$, 36$^0$, and 38$^0$) region during 2011, 2012, 2017, 2013–2015, 2018, and 2020. Similarly, Figure
11(d) depicts the gravity wave potential energy in the range of 2–30 J/kg for the lower MLT region
at 70 km over the longitudinal region (33-48$^0$). Out of which, the maximum potential energy of
25-30 J/kg is found in a certain longitude region over a while. Many of possible mechanisms have
been suggested for the cause of lower inversions; nonlinear interactions between GWs and tides
(Liu and Hagan, 1998), and chemical heating (Meriwether and Mlynczak, 1995) including GW
breaking (Hauchecorne et al., 1987). The role of gravity wave propagation and dissipation has
been accepted as the dominant wave forcing in the MLT region (Lindzen, 1981; Holton, 1983),
which affects the middle and upper atmospheric inversion. It is also understood that tides,
planetary waves, and chemical processes are affects the middle atmospheric variability as well as
gravity waves (Sivakandan et al., 2014). However, gravity waves are multi-scale in nature; small-
scale waves may contribute predominantly to instability, and turbulence in the MLT dynamic
region (Liu and Meriwether, 2004; Szewczyk et al., 2013).
Hence, investigating MIL phenomena is important for the understanding of MLT atmosphere
dynamics for two primary reasons: stability and energy transfer. As a result, an attempt has been
made to examine the contributions of gravity waves to the MLT region's instability (MIL
phenomenon) based on the Brunt-Vaisala frequency. The spatiotemporal variability of Vaisala
frequency is displayed in the contour Figure 12(a and b) for the upper mesosphere region (90 and
85 km) and Figure 12(c d) for the lower mesosphere region (75 and 70 km). Based on the
Brunt-Vaisala frequency, N2, the upper MLT region is unstable (~0.027) at 90 km and (~0.029) at
85 km maximum relative to the lower inversion instability at 75 km (~0.033) and 70 km (~0.035).
Hauchecorne et al., (1987) described a model in which a succession of breaking GWs would
generate the MIL through the gradual accumulation of heat as a cause of instability.

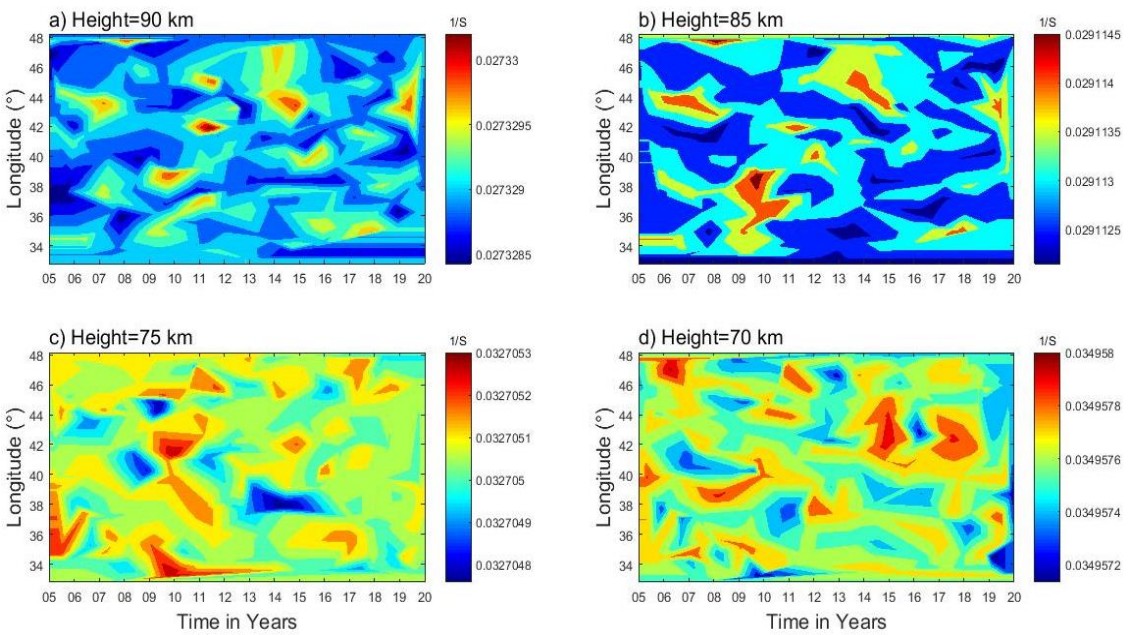


Figure 12. Brunt-Vaisala frequency ($N^2$) profiles for the upper (85 and 90 km) and lower (70 and
75 km) mesosphere regions.

## 4. Summary

In this article, 16 years of SABER mesosphere temperature profiles are utilized to investigate the MIL phenomenon and its causative mechanism through gravity wave potential energy ($P_E$) and instability criteria of Bruent-Vaisala frequency ($N^2$) over low latitude bands. The observational conclusions from this chapter are drawn as follows:

- ✓ The occurrence rate of the upper mesosphere inversion frequency is maximum relative to the mean occurrence rate of the lower mesosphere inversions.

- ✓ Based on the analysis of frequency of occurrence on mesospheric inversion layer (MIL) characteristic features, it is revealed that the most probable value for upper inversion amplitude is 38 k, inversion layer thickness is 5.5 km, and the base height is 78 km. Whereas the lower inversion amplitude is 25 K, the inversion layer thickness is 3.8 km, and a base height of 73 km.

- ✓ The gravity wave indicator potential energy depicts high energy at the upper mesosphere region compared to the lower mesosphere region.

- ✓ The result concludes that the observation of high potential energy in the upper mesosphere region is due to the deposition of high energy and momentum at the background temperature by gravity wave breaking, which could influence the dynamics of the inversion phenomenon

- ✓ The stability criteria at the mesosphere region are indicated by Brunt-Vaisala frequency ($N^2$), which shows low values at the upper mesosphere region relative to the lower mesosphere region, leading to the conclusion that the high potential energy at the upper mesosphere region is due to the instability over that region, which gives rise to large inversion phenomena.

- ✓ In general, we concluded that the processes in the atmosphere vary from region to region. As a result, the atmospheric state varies significantly with altitude as well as from place to place and time to time.

*Data availability.* The SABER data are freely available via the link at http://saber.gats-inc.com/index.php.

*Author contribution*. Chalachew Lingerew: data curation, investigation, software, visualization, writing the original draft, and writing review. U. Jaya Prakash Raju; supervision, and editing.

*Competing interest.* The authors declare that they have no conflict of interest relevant to this study.

*Acknowledgments.* The Authors would like to express their gratitude to the National Aeronautics and Space Administration (NASA) for providing the SABER data downloaded from the website: http://saber.gats-inc.com/index.php.

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
