# Peer review of "The Role of Gravity Waves in the Mesosphere-Lower-Thermosphere Inversion"

_Annales Geophysicae, 2023_

## Referee Comment (RC2)

Comments on, **"The Role of Gravity Waves in the Mesosphere Inversion Layers (MILs) over low latitude (3-15° N) Using SABER Satellite Observations"** by Lingerew and Raju.

Using sixteen years of SABER temperature data, the authors investigated the role of gravity waves (GWs) in the mesospheric inversion layers. To understand the role of GWs in the MILs, they estimate the potential energy, and based on the results they argue that the lower and upper MIL distinctions are due to the GWs. The strength of the manuscript is they used a long-term data set however their methodology is not clear. Moreover, this manuscript also lacks the scientific discussion. The present form of the manuscript needs major changes before acceptance for publication. Therefore, I recommend to the editor for a major revision. The detailed major and minor comments are as follows:

Major comments:
1. In section 2, latitudinal information of the data used is given however there is no information about the longitudes! Which reading the whole manuscript, I could see the longitudinal limits of 32 to 48° in Figures 10 and 11 (in section 3.4). Are the temperature profiles averaged over 3-15°N and 32-48°E? If so mention it in section 2. More importantly, the information about how do the MILs are identified is missing. They have only written as a diagnostic technique is used. What kind of diagnostic technique, whether the authors validated the diagnostic method all this information should be provided in the methodology (e.g. Gan et al., 2012; Sivakandan et al. 2014, etc.).

2. One of the major issues in the manuscript lack of a literature survey, though they have cited some of the important papers (Meriwether and Gardner 2000; Gan et al., 2012) but the essential points from those papers are not reflected in their approach. There are various sources proposed as the causative mechanism of the lower and upper MILs. For example, the planetary waves are believed to be the causative mechanism of lower MILs similarly, gravity wave tidal interactions and chemical heating are proposed as a cause of upper MILs. These points are not considered and there is no reason why the authors only focus on the GWs. It is well understood that in most of the cases the GWs breaking in the mesosphere can cause only very few Kelvin temperature changes (>10K). If this is the scenario it cannot explain the higher amplitude MILs. Comment on it.

3. As mentioned in comment 1, the authors should provide longitudinal information, because this has an important role if they try to understand the role of GWs which are highly localized in nature. It is not clear how the 1hr cutoff frequency applies to the data, if the authors used a particular region then in a day maximum of two to three satellite passes can be observed based on the area, with this limited data set how effective or logical is the 1hr bandpass filter?

4. Why 3rd order polynomial fit? Ramesh and Sridharan (2012) do not elaborate on any method, instead they have cited Leblanc and Hauchecorne (1997). Therefore the article cited here is not relevant. Provide more information about the methodology and its validity (how good it is? if the authors did any test to validate the method etc.)

5. Lines 138-140; in this context, Gan et al., (2012) could be a more suitable paper to cite here than Sivakumar et al. (2001), because they also used SABER data, on the other hand,

Sivakumar et al. (2001) only used Rayleigh lidar data over a single location (the data quality above 80 km is questionable). Gan et al. (2012) also found the seasonal variation of MILs in the low latitudes and planetary waves as the cause of lower MILs, whether these authors could find such a relationship? If yes or no provide reasons!

6. There is no clear information about how the occurrence frequency is estimated. Provide it?
7. How the mesopause altitudes are taken care or eliminated from the statistics? Which could be a false indication of inversion. And could the authors note any solar activity dependency of MILs occurrence (for example, Sivakandan et al. (2014))?
8. Lines 151-155, In the literature there are different causative mechanisms are proposed for the multiple MILs, (I suggest the authors go through Meriwether and Gardner (2004); Gan et al. (2012)).
9. Section 3.2, is a good point to investigate but before doing that the data need to be binned properly with local time. I am a bit concerned about how good to investigate the latitudinal and longitudinal variations in a small region using satellite data, each temperature profile could be nearly 500 km spatial averaged.
10. The scientific discussion is very spare and weak. They should compare the present results with earlier studies based on the similarities and differences the scientific reasoning also should be included in the manuscript.
11. How the GWs potential energy is connected to the MILs? First establish the connection by showing a single case in which a physical connection should be clear and then go for the statistics.

Minor comments:
12. Lines 8-9, The mesosphere…This is a transitional region not only in the low latitudes! So modify the statement.
13. Lines 39-40, define the MILs.
14. Line 41, a typo, 'mesosphere'
15. Line 73, these references are irrelevant here. Provides references about the data validation and limitation as well as instrumental specifications.
16. Line 75, longitudinal information is missing!
17. Figure 4: Sivakandan et al. (2014) also did such a statistical analysis using the SABER data over Indian low latitudes, could you compare the present results with their results and provide some scientific reasoning for the observed differences or similarities?
18. Line 218 …that the inversion temperature is in the range of…It is not an inversion temperature range only a temperature range.
19. Line 242 onwards, the longitudinal information is suddenly introduced here, it should be introduced in section 2.
20. Lines 245-247, these lines are not clear. Please see the major comment 3.
21. Figure 5b, a typo 'thickness'

References suggested to read and compare with the present results and include in the discussion part (some of the articles are cited here but those results are not utilized to improve the discussion part):

1. Gan, Q., S. D. Zhang, and F. Yi (2012), TIMED/SABER observations of lower mesospheric inversion layers at low and middle latitudes, *J. Geophys. Res.*, 117, D07109, doi:10.1029/2012JD017455.
2. Meriwether, J. W., and C. S. Gardner (2000), A review of the mesosphere inversion layer phenomenon, *J. Geophys. Res.*, 105(D10), 12405–12416, doi:10.1029/2000JD900163.
3. Sivakandan, M., Kapasi, D., and Taori, A.: The occurrence altitudes of middle atmospheric temperature inversions and mesopause over low-latitude Indian sector, Ann. Geophys., 32, 967–974, https://doi.org/10.5194/angeo-32-967-2014, 2014.
4. Ramesh, K., S. Sridharan, and S. Vijaya Bhaskara Rao (2014), Causative mechanisms for the occurrence of a triple layered mesospheric inversion event over low latitudes, *J. Geophys. Res. Space Physics*, 119, 3930–3943, doi:10.1002/2013JA019750.

---

## Author Comment (AC4)

Comments on, **"The Role of Gravity Waves in the Mesosphere Inversion Layers (MILs) over low latitude (3-15° N) Using SABER Satellite Observations"** by Lingerew and Raju. Using sixteen years of SABER temperature data, the authors investigated the role of gravity waves (GWs) in the mesospheric inversion layers. To understand the role of GWs in the MILs, they estimate the potential energy, and based on the results they argue that the lower and upper MIL distinctions are due to the GWs. The strength of the manuscript is they used a long-term data set however their methodology is not clear. Moreover, this manuscript also lacks the scientific discussion. The present form of the manuscript needs major changes before acceptance for publication. Therefore, I recommend to the editor for a major revision. The detailed major and minor comments are as follows:

**Major comments:**

1. In section 2, latitudinal information of the data used is given however there is no information about the longitudes! Which reading the whole manuscript, I could see the longitudinal limits of 32 to 48° in Figures 10 and 11 (in section 3.4). Are the temperature profiles averaged over 3-15°N and 32-48°E? If so mention it in section 2. More importantly, the information about how do the MILs are identified is missing. They have only written as a diagnostic technique is used. What kind of diagnostic technique, whether the authors validated the diagnostic method all this information should be provided in the methodology (e.g. Gan et al., 2012; Sivakandan et al. 2014, etc.).

Response: We have mentioned the criteria to separate the inversion phenomenon from the observation data in lines 87-91 from Section 2.

2. One of the major issues in the manuscript lack of a literature survey, though they have cited some of the important papers (Meriwether and Gardner 2000; Gan et al., 2012) but the essential points from those papers are not reflected in their approach. There are various sources proposed as the causative mechanism of the lower and upper MILs. For example, the planetary waves are believed to be the causative mechanism of lower MILs similarly, gravity wave tidal interactions and chemical heating are proposed as a cause of upper MILs. These points are not considered and there is no reason why the authors only focus on the GWs. It is well understood that in most of the cases the GWs breaking in the mesosphere can cause only very few Kelvin temperature changes (>10K). If this is the scenario it cannot explain the higher amplitude MILs. Comment on it.

Response: OK, thank you. We know that all tidal, planetary, and gravity waves, as well as chemical reactions, are causative of an inversion, but here in our study region between 60 and 90 km is the gravity wave, which is generated from the lower atmosphere and propagated to the upper atmosphere till to reach the saturation level for breaking and it impacts the atmospheric variability as causative of an inversion.

3. As mentioned in comment 1, the authors should provide longitudinal information, because this has an important role if they try to understand the role of GWs which are highly localized in nature. It is not clear how the 1hr cutoff frequency applies to the data, if the authors used a particular region then in a day maximum of two to three satellite passes can

be observed based on the area, with this limited data set how effective or logical is the 1hr band pass filter?

Response: we have used only SABER temperature data over the low-latitude in the spatial regions from (3-15) latitude, (32-48) longitude and (60-100 km) altitude. So, as mentioned in the text, we have applied a one-hour interval cut-off frequency of the pass band filter to separate the gravity waves from those other wave activities, such as planetary and tidal atmospheric waves.

4. Why 3rd order polynomial fit? Ramesh and Sridharan (2012) do not elaborate on any method, instead they have cited Leblanc and Hauchecorne (1997). Therefore the article cited here is not relevant. Provide more information about the methodology and its validity (how good it is? if the authors did any test to validate the method etc.)

Response: OK, we can use the reference Leblanc and Hauchecorne (1997) instead of Ramesh and Sridharan (2012) based on their relevance to express the third-order polynomial fit. The 3rd-order polynomial fit is relatively good, as we understand from the scientific community, to provide the background information relative to other orders. Then after having this information we can derive the perturbations by subtracting the background from the observation data.

5. Lines 138-140; in this context, Gan et al., (2012) could be a more suitable paper to cite here than Sivakumar et al. (2001), because they also used SABER data, on the other hand, Sivakumar et al. (2001) only used Rayleigh lidar data over a single location (the data quality above 80 km is questionable). Gan et al. (2012) also found the seasonal variation of MILs in the low latitudes and planetary waves as the cause of lower MILs, whether these authors could find such a relationship? If yes or no provide reasons!

Response: It doesn't need any reason for yes or no to use the reference (Gan et al., 2012) instead of (Sivakumar et al., 2001), because already you have mentioned why Gan is preferable to Sivakumar based on their relevance to supporting the idea about the base of the lower inversion in lines 138–140 so we simply accepted using that one.

6. There is no clear information about how the occurrence frequency is estimated. Provide it?

Response: We can calculate the occurrence rate (percentages) for lower and upper inversions by counting the number of inversion days every month from 2005 to 2020.

7. How the mesopause altitudes are taken care or eliminated from the statistics? Which could be a false indication of inversion. And could the authors note any solar activity dependency of MILs occurrence (for example, Sivakandan et al. (2014))?
Response: We did our work about inversions and their causative gravity waves in MLT dynamic regions over low latitudes, but we didn't consider the pous (mesopous).

8. Lines 151-155, In the literature there are different causative mechanisms are proposed for the multiple MILs, (I suggest the authors go through Meriwether and Gardner (2004); Gan et al. (2012)).

Response: OK, we will check again on their scientific investigations.

9. Section 3.2, is a good point to investigate but before doing that the data need to be binned properly with local time. I am a bit concerned about how good to investigating the latitudinal and longitudinal variations in a small region using satellite data, each temperature profile could be nearly 500 km spatial averaged.

Response: It is not local time; instead, we have used the period during 2005–2020 over the latitudinal regions (3–15) and longitudinal regions (32-48).

10. The scientific discussion is very spare and weak. They should compare the present results with earlier studies based on the similarities and differences the scientific reasoning also should be included in the manuscript.

Response: OK, we will try to elaborate the discussion based on your comment.

11. How the GWs potential energy is connected to the MILs? First establish the connection by showing a single case in which a physical connection should be clear and then go for the statistics.

Response: Relay, we have background information on how gravity waves impact the inversions of the upper atmosphere. The connection is that a gravity wave is generated from the lower atmosphere, and the wave propagates to the upper region until it reaches the saturation level over the upper region. The wave is then broken to dissipate the energy, and its energy impacts the region by increasing its temperature. In this region, the temperature increment with elevation is known as an inversion. This is the reason we connected the gravity waves with an inversion.

**Minor comments:**

12. Lines 8-9, The mesosphere…This is a transitional region not only in the low latitudes! So modify the statement.

Response: The Mesosphere transitional region over low latitudes is a distinct and highly turbulent zone of the atmosphere relative to mid- and high latitudes.

13. Lines 39-40, define the MILs.

Response: The Mesospheric Inversion Layer (MIL) is a feature that increases the temperature profiles in the mesosphere region.

14. Line 41, a typo, 'mesosphere'

Response: OK, we corrected

15. Line 73, these references are irrelevant here. Provides references about the data validation and limitation as well as instrumental specifications.

Response:

16. Line 75, longitudinal information is missing!

Response: OK, we will correct it.

17. Figure 4: Sivakandan et al. (2014) also did such a statistical analysis using the SABER data over Indian low latitudes, could you compare the present results with their results and provide some scientific reasoning for the observed differences or similarities?

Response: OK, thank you. We will try to check and compare with his scientific results.

18. Line 218 …that the inversion temperature is in the range of…It is not an inversion temperature range only a temperature range.

Response: Yes, it is the inversion-day observed temperature.

19. Line 242 onwards, the longitudinal information is suddenly introduced here, it should be introduced in section 2.

Response: Ok

20. Lines 245-247, these lines are not clear. Please see the major comment 3.

Response: OK, we will elaborate

21. Figure 5b, a typo 'thickness' References suggested to read and compare with the present results and include in the discussion part (some of the articles are cited here but those results are not utilized to improve the discussion part):

Response: It is corrected and includes the references based on their relevance.

1. Gan, Q., S. D. Zhang, and F. Yi (2012), TIMED/SABER observations of lower mesospheric inversion layers at low and middle latitudes, *J. Geophys. Res.*, 117, D07109, doi:10.1029/2012JD017455.
2. Meriwether, J. W., and C. S. Gardner (2000), A review of the mesosphere inversion layer phenomenon, *J. Geophys. Res.*, 105(D10), 12405–12416, doi:10.1029/2000JD900163.
3. Sivakandan, M., Kapasi, D., and Taori, A.: The occurrence altitudes of middle atmospheric temperature inversions and mesopause over low-latitude Indian sector, Ann. Geophys., 32, 967–974, https://doi.org/10.5194/angeo-32-967-2014, 2014.
4. Ramesh, K., S. Sridharan, and S. Vijaya Bhaskara Rao (2014), Causative mechanisms for the occurrence of a triple layered mesospheric inversion event over low latitudes, *J. Geophys. Res. Space Physics*, 119, 3930–3943, doi:10.1002/2013JA019750.

---

## Author Response (AR1)

**We would like to express our gratitude for the valuable feedback from the Referee. We appreciate your tireless efforts in reviewing our manuscript so it allows us to improve the standard of our article.**

**RC1**

This work is devoted to an interesting and insufficiently studied topic. Results of variations in mesospheric inversion layers are presented, but there are some comments:

1. Line 41: missing letter «e» in mesosphere

   Response: We have corrected the typographical error "mesospher" by mesosphere in line 40.

2. Line 74-77: The sentence is too long. And Figure 1 shows temperature, not causatives. Please rephrase.

   Response: We have modified and corrected the sentence in lines 80-84 in the new revision as well as corrected the expressions of Figure 1.

3. Figure 2 (as well as Figure 7.8) is not entirely clear. There are too many black isolines and the fill behind them is not visible. Try increasing the size of Figures and thinning out the isolines. Maybe leave one isoline that satisfies the inversion criterion.

   Response: We have enlarged Figures 2, 7, and 8 for visibility as well as rewritten the sentences for clarity.

4. Line 140-141: You claim that the base of the lower MILs lies in the range of 73-79, the upper MILs is 86-89 km. Sorry, but I don't see this in Figure 2. Why did you indicated these heights?

   Response: We have corrected the sentences in lines 178–18, and we can see the corresponding Figure 2.

5. I don't understand why histogram 4(c) shows a maximum MILs at 78 km, but histogram 4(f) does not have a maximum at this altitude.

   Response: Thank you. Figure 4(c and f) shows the number of upper and lower inversions in the given intervals of base height ~ (70–80) for the lower MLT region and base height ~ (80–90) for the upper MLT region. Figures 4(c) and 4(f) are corrected in the revised manuscripts.

6. In my opinion, the conclusions are overloaded with numbers. All parameters are listed in the text of the manuscript.

   Response: Thank you for your valuable suggestion. The conclusion has been modified based on your suggestion.

7. To what extent is it physically justified to explain the MILs by changes in the Brunt-Vaisala frequency? After all, in essence these are the same things; if there is an inversion, it means the atmosphere is not stable.

   Response: In this investigation, the Brunt-Vaisala frequency is only used to show the instabilities in the Mesospheric inversion layers (MILs).

8. The manuscript also lacks interpretation of physical processes. There is no attempt to speculate about the sources of gravity waves. Although Figure 1 clearly shows a quasiperiodic structure of temperature fluctuations. The periodicity is also visible in Figure 3. How does this relate to the QBO phase? Or with other processes in the low-latitude atmosphere. Perhaps winds and shears in the mesosphere generated gravity waves. Your figures show significant interannual variability. What features were there in 2012-2013 and 2019? Gravity wave potential energy is increased at longitudes 35-40E, maybe the source was the lower layers of the atmosphere, for example, The Low-Level Somali Jet? Discussions about the physical nature of the origin of gravity waves should be added to the manuscript.

   Response: We thank you for your valuable suggestion. After carefully considering your recommendations we tried to modify the interpretation of physical processes even the source of the gravity waves and their impacts are clearly stated in the manuscript. We can see section 3.4.

**RC2**

Comments on, **"The Role of Gravity Waves in the Mesosphere Inversion Layers (MILs) over low latitude (3-15° N) Using SABER Satellite Observations"** by Lingerew and Raju. Using sixteen years of SABER temperature data, the authors investigated the role of gravity waves (GWs) in the mesospheric inversion layers. To understand the role of GWs in the MILs, they estimate the potential energy, and based on the results they argue that the lower and upper MIL distinctions are due to the GWs. The strength of the manuscript is they used a long-term data set however their methodology is not clear. Moreover, this manuscript also lacks the scientific discussion. The present form of the manuscript needs major changes before acceptance for publication. Therefore, I recommend to the editor for a major revision. The detailed major and minor comments are as follows:

**Major comments:**

1. In section 2, latitudinal information of the data used is given however there is no information about the longitudes! Which reading the whole manuscript, I could see the longitudinal limits of 32 to 48° in Figures 10 and 11 (in section 3.4). Are the temperature profiles averaged over 3-15°N and 32-48°E? If so mention it in section 2. More importantly, the information about how do the MILs are identified is missing. They have only written as a diagnostic technique is used. What kind of diagnostic technique, whether the authors validated the diagnostic method all this information should be provided in the methodology (e.g. Gan et al., 2012; Sivakandan et al. 2014, etc.).

Response: Thank you for your comment. Based on your comment, we have mentioned the criteria to separate the inversion phenomenon from the observation data in lines 92–106 by including the references you mentioned in Section 2 from lines 101-104, as well as adding the longitudinal information in line 82.

2. One of the major issues in the manuscript lack of a literature survey, though they have cited some of the important papers (Meriwether and Gardner 2000; Gan et al., 2012) but the essential points from those papers are not reflected in their approach. There are various sources proposed as the causative mechanism of the lower and upper MILs. For example, the planetary waves are believed to be the causative mechanism of lower MILs similarly, gravity wave tidal interactions and chemical heating are proposed as a cause of upper MILs. These points are not considered and there is no reason why the authors only focus on the GWs. It is well understood that in most of the cases the GWs breaking in the mesosphere can cause only very few Kelvin temperature changes (>10K). If this is the scenario it cannot explain the higher amplitude MILs. Comment on it.

Response: OK, thank you. all tidal, planetary, and gravity waves, as well as chemical reactions, are causative of an inversion, but here in our study region between 60 and 90 km, only the gravity wave is considered, which is generated from the lower atmosphere and propagated to the upper atmosphere till it reaches the saturation level for breaking and impacts the atmospheric variability as causative of an inversion.

3. As mentioned in comment 1, the authors should provide longitudinal information, because this has an important role if they try to understand the role of GWs which are highly localized in nature. It is not clear how the 1hr cutoff frequency applies to the data, if the authors used a particular region then in a day maximum of two to three satellite passes can be observed based on the area, with this limited data set how effective or logical is the 1hr band pass filter?

Response: we have used only SABER temperature data over the spatial regions of latitude (3–15), longitude (32-48), and altitude (60–90 km) during 2005-2020. So, as mentioned in the text, we have applied a one-hour interval cut-off frequency of the low-pass band filter to separate the

gravity waves from those other wave activities, such as planetary and tidal atmospheric wave impacts. For more clarity, we can see the explanations of the techniques of how to use the low pass band filters, which have a 1-h cut-off frequency to separate the short periodic gravity waves, from tidal and planetary waves in lines 148-153.

4. Why 3rd order polynomial fit? Ramesh and Sridharan (2012) do not elaborate on any method, instead they have cited Leblanc and Hauchecorne (1997). Therefore the article cited here is not relevant. Provide more information about the methodology and its validity (how good it is? if the authors did any test to validate the method etc.)

Response: The scientific community prefers the 3rd-order polynomial fit to estimate the background information relative to other orders (Leblanc and Hauchecorne, 1997; Ramesh and Sridharan, 2012).

5. Lines 138-140; in this context, Gan et al., (2012) could be a more suitable paper to cite here than Sivakumar et al. (2001), because they also used SABER data, on the other hand, Sivakumar et al. (2001) only used Rayleigh lidar data over a single location (the data quality above 80 km is questionable). Gan et al. (2012) also found the seasonal variation of MILs in the low latitudes and planetary waves as the cause of lower MILs, whether these authors could find such a relationship? If yes or no provide reasons!

Response: We have used the relevant references to support our findings about the base of the lower inversions as presented in lines 178–182.

6. There is no clear information about how the occurrence frequency is estimated. Provide it?

Response: The occurrence rate (percentages) for lower and upper inversions is estimated by counting the number of inversion days every month from 2005 to 2020.

7. How the mesopause altitudes are taken care or eliminated from the statistics? Which could be a false indication of inversion. And could the authors note any solar activity dependency of MILs occurrence (for example, Sivakandan et al. (2014))?
Response: We did our work about inversions and their causative gravity waves in upper and lower MLT dynamic regions over low latitudes, but we didn't consider the mesopous.

8. Lines 151-155, In the literature there are different causative mechanisms are proposed for the multiple MILs, (I suggest the authors go through Meriwether and Gardner (2004); Gan et al. (2012)).

Response: We have different causative atmospheric waves for an inversion, such as planetary waves, tidal waves, and gravity waves, as well as chemical reactions, but here we are considering only gravity waves as an impact on an inversion.

9. Section 3.2, is a good point to investigate but before doing that the data need to be binned properly with local time. I am a bit concerned about how good to investigating the latitudinal

and longitudinal variations in a small region using satellite data, each temperature profile could be nearly 500 km spatial averaged.

Response: Thank you for your comment. However, it is not a local time investigation; instead, we have used the period from 2005 to 2020 over the latitudinal (3–15) and longitudinal (32-48) spatial regions.

10. The scientific discussion is very spare and weak. They should compare the present results with earlier studies based on the similarities and differences the scientific reasoning also should be included in the manuscript.

Response: Thank you for your suggestion to improve our manuscripts. Hence, based on your comment, whole sections of the manuscript were developed by rewriting the scientific discussion in the revision.

11. How the GWs potential energy is connected to the MILs? First establish the connection by showing a single case in which a physical connection should be clear and then go for the statistics.

Response: Thank you. We have presented their connections in Section 3.4, particularly in lines 310–313, as follows: "The saturation stage of the wave propagation is broken at the upper region to dissipate the energy, which impacts the normal mesospheric temperature by increasing its temperature with elevation, known as an inversion. This is the reason the gravity wave potential energy is connected with an inversion."

**Minor comments:**

12. Lines 8-9, The mesosphere…This is a transitional region not only in the low latitudes! So modify the statement.

Response: The sentence you mentioned in lines 8–9 is modified as follows: "The Mesosphere transitional region is a distinct and highly turbulent zone of the atmosphere."

13. Lines 39-40, define the MILs.

Response: OK, thank you. We have modified the inversions, MILs, in lines 38–40 as follows: "The mesospheric inversion layers (MILs) are a common feature that appeared to increase the mesosphere temperature variability.

14. Line 41, a typo, 'mesosphere'

Response: OK, this typo error is corrected

15. Line 73, these references are irrelevant here. Provides references about the data validation and limitation as well as instrumental specifications.

Response: We have added the references to express the validation and the limitations of SABER observations in the MLT region in lines 75-79.

16. Line 75, longitudinal information is missing!

Response: Ok now it is corrected in the revised manuscript from line 82.

17. Figure 4: Sivakandan et al. (2014) also did such a statistical analysis using the SABER data over Indian low latitudes, could you compare the present results with their results and provide some scientific reasoning for the observed differences or similarities?

Response: Yes really we have seen his scientific work results as a comparison with our work results and we mention their difference and similarity based on their statistical analysis in the newly revised manuscript from section 3.1.

18. Line 218 …that the inversion temperature is in the range of…It is not an inversion temperature range only a temperature range.

Response: It is the inversion-day observed temperature.

19. Line 242 onwards, the longitudinal information is suddenly introduced here, it should be introduced in section 2.

Response: Now it is corrected by introducing the longitudinal information in section 2.

20. Lines 245-247, these lines are not clear. Please see the major comment 3.

Response: It is corrected by rewriting the sentences in lines 316-326

21. Figure 5b, a typo 'thickness' References suggested to read and compare with the present results and include in the discussion part (some of the articles are cited here but those results are not utilized to improve the discussion part):

Response: We have corrected the typo error and included the references you have mentioned below based on their relevance.

1. Gan, Q., S. D. Zhang, and F. Yi (2012), TIMED/SABER observations of lower mesospheric inversion layers at low and middle latitudes, *J. Geophys. Res.*, 117, D07109, doi:10.1029/2012JD017455.
2. Meriwether, J. W., and C. S. Gardner (2000), A review of the mesosphere inversion layer phenomenon, *J. Geophys. Res.*, 105(D10), 12405–12416, doi:10.1029/2000JD900163.
3. Sivakandan, M., Kapasi, D., and Taori, A.: The occurrence altitudes of middle atmospheric temperature inversions and mesopause over low-latitude Indian sector, Ann. Geophys., 32, 967–974,https://doi.org/10.5194/angeo-32-967-2014,2014.
4. Ramesh, K., S. Sridharan, and S. Vijaya Bhaskara Rao (2014), Causative mechanisms for the occurrence of a triple layered mesospheric inversion event over low latitudes, *J. Geophys. Res. Space Physics*, 119, 3930–3943, doi:10.1002/2013JA019750.

**RC3**

The manuscript presents an investigation of the likely effects of atmospheric gravity waves on Mesospheric Inversion Layers (MILs) in equatorial latitudes. Indeed, the topic is interesting and not explored very well by the communities. Additionally, it is within the scope of Annales Geophysicae because it is an experimental investigation of the mesosphere using satellite measurements. Before I consider the paper suitable for publication, some concerns could be explored and revised by the authors to improve the quality of the manuscript. In general, the observations present in all figures were not explained or explored very well. Consequently, I missed consistent interpretations of the present results.

1. For example, Figure 1 shows the variability of the temperature in the mesosphere and lower thermosphere, what is the objective of it? Is it possible to see MILs, where? If not, why is it not possible to see? In summary, how can Figure 1 help the authors?

   Response: Figure 1 presents the observed temperature variability of the MLT region before segregating the inversions.

2. Figure 2 shows upper (left) and lower (right) MILs, to be sincere, I did not understand the bottom panels. May the authors explain them better? I also suggest enlarging captions and the size of the panels.

   Response: The upper panel of Figure 2 (a) and (c) represents the upper and lower MLT SABER observed temperatures whereas the lower panel of Figure 2 (b) and (d) represent the upper and lower inversions of the MLT temperature variability, respectively, which is after separated from Figure 2 (a & c). The figure is now enlarged.

3. Figure 3: what is the bin size and which criteria were used to determine the percentage of occurrence of MILs?

   Response: The occurrence rate (percentages) for lower and upper inversions is estimated by counting the number of inversion days for every month from 2005 to 2020.

4. Figure 4: Are the red curves indicating a Gaussian fit? If yes, please, explain it in the text. Please, note that the statistic used for panel (c) does not represent the data, in this case, the authors could use another statistic or explain what are causing the discrepancies.

   Response: The red curve fitting is the Gaussian fit distribution, which is a probability distribution that is symmetric about the mean, showing that data near the mean are more frequent in occurrence than data far from the mean, which is mentioned in lines 199–205. I

have mentioned their statistical values by corrected the Figure 4(c) & 4(f) in the revised manuscripts.

5. Figure 5: What is the relevance of these results and how could they be related to gravity waves? I guess the quality of the presentation of this figure could be improved by enlarging the caption and size. Figure 6: Same comments as Figure 5.2.

Response: Before investigating the impacts of gravity waves on the MLT inversions, the spatiotemporal (time vs. latitude) variabilities of an inversion are presented in Figures 5 and 6 to characterize the upper and lower inversions, respectively. The figure is now enlarged for clarity.

6. I recommend the authors use a more complete expression of the potential temperature (Vadas and Fritts, 2005; Vdas, 2007) instead of what was presented in Equation (1). In addition, the discussion of atmospheric stability is very superficial and it does not include a real aspect of the atmosphere that is certainly present in the SABER data. I suggest including an example of the methodology used to calculate the MILs in the real data (section 2). In my opinion, it could help the readers to promptly understand the process.

Response: Thank you for your valuable suggestion. Based on your recommendation, we have used the references [(Liu, 2011; Vadas and Fritts, 2005); (e.g. Gan et al., 2012 and Sivakandan et al. 2014)] along with equations 1–13 to elaborate on the expressions of the gravity wave potential energy instead of using a single equation, as well as calculating the Brunt-Vaisala frequency, $N^2$, to characterize atmospheric stability in the methodology part of Section 2.2 for more clarification for readers.

7. Section 3.4: The authors wrote that they used a low pass filter to exclude the effects of tidal and planetary waves in the residual signal. If I understood the process, I guess they could use high-frequency filters to maintain low periods. Indeed, the blue lines in Figure 9 show a smoothed signal that excludes short-time variations. If I am correct, Figures 10 and 11 could be revised and the interpretations as well.

Response: we have used a 1-h interval cut-off frequency of a low-pass band filter, in which Figure 9 shows smooth data signals by removing the peak while using 1-h interval perturbed temperature data profiles to exclude more than a 1-h time variation. The discussion part of the results in Figures 10 and 11 is nearly rewritten.

8. Lines 279-282: "The result concludes that the observation of high potential energy in the upper mesosphere region is due to the deposition of high energy and momentum at the background temperature by gravity wave breaking, which could influence the dynamics of the inversion phenomenon". Lines 291-293: "This result leads us to the conclusion that a high amount of gravity wave potential energy is a consequence of the high instability of the upper inversion relative to the lower." I guess, it is possible to reach these conclusions from the present work. Conclusions: I think it will be better to change the name of the section to Summary and exclude the last two ones that are very general.

Response: The sentences are removed from the sections you mentioned in lines 279–282 and 291-293; instead, they are included in the summary section in lines 391–395. The subtitle, Section 4, conclusion, is changed to summary. The references you commented on are included in the revised manuscript.

**Reference**

1. Garcia-Comas, M., Lopez-Puertas, M., Marshall, B. T., Wintersteiner, P. P., Funke, B., Bermejo-Pantaleon, D., Mertens, C. J., Remsberg, E. E., Gordley, L. L., Mlynczak, M. G., and Russell III, J. M.: Errors in Sounding of the Atmosphere using Broadband Emission Radiometry (SABER) kinetic temperature caused by non-local-thermodynamic-equilibrium model parameters, J. Geophys. Res., 113, D24106, doi:10.1029/2008JD010105, 2008.
2. Meriwether, J. W., and Gerrard, A. J.: Mesosphere inversion layers and stratosphere temperature enhancements, Rev. Geophys., 42, RG3003, http//doi:10.1029/2003RG000133, 2004.

---

## Referee Report (RR1)

Thanks to the authors for the answers, the paper has become a little better. However, there are still notes:

Please clarify, is this MIL?

[Figure]

2. I already recommended making Figure 3 clear and understandable. How should I understand it? Everything is in black lines and motley spots. How to evaluate the heights that you give in description? How to understand where there is a temperature inversion day and where there is not? Please remove the black outline and leave just the fill.

3. You are considering MIL over 3-15 N latitude and 33-48 E longitude regions. Why did you choose this region? Write a few words about this. It's better to put the coordinates of the area in the title, otherwise it seems that you are doing research over all low latitudes. As a last resort, add it to the abstract on line 17.

4. I didn't understand your idea, what does «….seasonal variations of MIL in the planetary waves as a caustive over low-latitudes using the SABER observations» mean? And there is a typo " caustive".

5. And now in Figures 5(c) and 5(f) the averages are on top of the bars. Please, correct it.

6. The last conclusion sounds strange and vague - « In general, we concluded that the processes in the atmosphere vary from region to region. As a result, the atmospheric state varies significantly with altitude as well as from place to place and time to time»; I guess that this conclusion should be removed.

---

## Author Response (AR2)

We would like to express our gratitude for the valuable feedback from the Referee. We appreciate your tireless efforts in reviewing our manuscript so it allows us to improve the standard of our article.

**RC1**

This work is devoted to an interesting and insufficiently studied topic. Results of variations in mesospheric inversion layers are presented, but there are some comments:

1. Line 41: missing letter «e» in mesosphere

   Response: We have corrected the typographical error "mesospher" to "mesosphere" in line 40.

2. Line 74-77: The sentence is too long. And Figure 1 shows temperature, not causatives. Please rephrase.

   Response: We have modified to correct the long sentence in lines 84–88 in the new revision, as shown below. Additionally, we have corrected the expressions in Figure 1, which now display SABER observed MLT temperature instead of causative.

   "In the present, we have used the latest version of SABER temperature data over low-latitudes. The SABER vertical temperature profiles were taken in the range of 60-100 km altitude during the period January 2005-December 2020 over $(3\text{-}15^0\text{N})$ latitude and $(33\text{-}48^0\text{E})$ longitude regions. The monthly mean SABER temperature data of the mesosphere and lower thermosphere (MLT) region is presented as shown in Figure 1."

3. Figure 2 (as well as Figure 7.8) is not entirely clear. There are too many black isolines and the fill behind them is not visible. Try increasing the size of Figures and thinning out the isolines. Maybe leave one isoline that satisfies the inversion criterion.

   Response: Thank you for your comment. Based on your suggestions for visibility, we have enlarged and clarified Figures 3, 8, and 9.

4. Line 140-141: You claim that the base of the lower MILs lies in the range of 73-79, and the upper MILs is 86-89 km. Sorry, but I don't see this in Figure 2. Why did you indicated these heights?

   Response: We have corrected the sentences based on the output from Figure 3 in lines 195–199, and we can now observe the corresponding Figure 3.

"Our findings of the lower inversions in the range of (70-80 km) tend to approach the reports by Sivakumar et al. (2001), which show that the base of the lower mesospheric inversion layer (MILs) lies in the range of (73-79 km), as well as the Gan et al. (2012) seasonal variations of MIL in the planetary waves as a caustive over low-latitudes using the SABER observations."

5. I don't understand why histogram 4(c) shows a maximum MILs at 78 km, but histogram 4(f) does not have a maximum at this altitude.

Response: Thank you for your comment. Figures 5(c) and 5(f) have been corrected in the revised manuscript. Figure 5(c) shows the upper inversion, while Figure 5(f) shows the lower inversion, in the specified intervals of base height (~70–80) and base height (~80–90) for the upper MLT region.

In lines 238-240, "The base height of the upper MIL in Figure 5(c) ranges from ~80 to 90 km, with a peak value of a large number of upper mesospheric inversions occurring at a base height of around 83 km in a lower standard deviation (SD) 2.13."

In lines 248-250, "The base height of the lower inversion of Figure 5(f) is in the range of 70 and 80 km, with a peak value of around 74 km, showing a lower standard deviation (SD) of 1.93."

6. In my opinion, the conclusions are overloaded with numbers. All parameters are listed in the text of the manuscript.

Response: Thank you for your valuable suggestion. The conclusion has now been modified based on your suggestion in section 4, from lines 395-418, by reducing the number of crowded details as shown below.

**"3. Summary**

In this article, 16 years of SABER mesosphere temperature profiles are utilized to investigate the MIL phenomenon and its causative mechanism through gravity wave potential energy ($P_E$) and instability criteria of Bruent-Vaisala frequency ($N^2$) over low latitude bands. The observational conclusions from this chapter are drawn as follows:

- ✓ The occurrence rate of the upper mesosphere inversion frequency is maximum relative to the mean occurrence rate of the lower mesosphere inversions.
- ✓ Based on the analysis of frequency of occurrence on mesospheric inversion layer (MIL) characteristic features, it is revealed that the most probable value for upper inversion amplitude is 38 k, inversion layer thickness is 5.5 km, and the base height is 78 km. Whereas the lower inversion amplitude is 25 K, the inversion layer thickness is 3.8 km, and a base height of 73 km.
- ✓ The gravity wave indicator potential energy depicts high energy at the upper mesosphere region compared to the lower mesosphere region.
- ✓ The result concludes that the observation of high potential energy in the upper mesosphere region is due to the deposition of high energy and momentum at the background temperature by gravity wave breaking, which could influence the dynamics of the inversion phenomenon
- ✓ The stability criteria at the mesosphere region are indicated by Brunt-Vaisala frequency ($N^2$), which shows low values at the upper mesosphere region relative to the lower mesosphere region, leading to the conclusion that the high potential energy at the upper mesosphere region is due to the instability over that region, which gives rise to large inversion phenomena.
- ✓ In general, we concluded that the processes in the atmosphere vary from region to region. As a result, the atmospheric state varies significantly with altitude as well as from place to place and time to time."

7. To what extent is it physically justified to explain the MILs by changes in the Brunt-Vaisala frequency? After all, in essence these are the same things; if there is an inversion, it means the atmosphere is not stable.

Response: Thank you for your suggestion and comment. In this investigation, the Brunt-Vaisala frequency is used to demonstrate the instabilities of the Mesospheric Inversion Layers (MILs). Our aim here is to assess by which extents of gravity waves impact atmospheric instability, based on the Brunt-Vaisala formulation, as explained in Figure 11 of the manuscript (lines 382-391), as described in lines 334-336.

"In this section, an attempt has been made to investigate the longitudinal variability of gravity waves' contribution to the mesospheric inversions (MILs) phenomenon by calculating potential energy and their instability based on Bruent-Vaisala frequency ($N^2$) using perturbed temperatures."

"Hence, investigating MIL phenomena is important for the understanding of MLT atmosphere dynamics for two primary reasons: stability and energy transfer. As a result, an attempt has been made to examine the contributions of gravity waves to the MLT region's instability (MIL phenomenon) based on the Brunt-Vaisala frequency. The spatiotemporal variability of Vaisala frequency is displayed in the contour Figure 11(a and b) for the upper mesosphere region (90 and 85 km) and Figure 11(c and d) for the lower mesosphere region (75 and 70 km). Based on the Brunt-Vaisala frequency, N2, the upper MLT region is unstable (~0.027) at 90 km and (~0.029) at 85 km maximum relative to the lower inversion instability at 75 km (~0.033) and 70 km (~0.035). Hauchecorne et al., (1987) described a model in which a succession of breaking GWs would generate the MIL through the gradual accumulation of heat as a cause of instability."

8. The manuscript also lacks interpretation of physical processes. There is no attempt to speculate about the sources of gravity waves. Although Figure 1 clearly shows a quasiperiodic structure of temperature fluctuations. The periodicity is also visible in Figure 3. How does this relate to the QBO phase? Or with other processes in the low-latitude atmosphere. Perhaps winds and shears in the mesosphere generated gravity waves. Your figures show significant interannual variability. What features were there in 2012-2013 and 2019? Gravity wave potential energy is increased at longitudes 35-40E, maybe the source was the lower layers of the atmosphere, for example, The Low-Level Somali Jet? Discussions about the physical nature of the origin of gravity waves should be added to the manuscript.

Response: We thank you for your valuable comment and suggestion. Based on your comment we modified the interpretation of physical processes even the source of the gravity waves and their impacts are clearly stated in the manuscript as follows in lines 319-330.

"Atmospheric waves (gravity waves, planetary waves, and tidal waves) exist in different layers of the atmosphere and are generated by different mechanisms. Gravity waves are of local or regional dimensions, whereas the other two waves are of global extent. This dynamical gravity wave motion is a restoring force of gravity acting downward and buoyancy acting upward on vertically displaced air parcels from the troposphere/stratosphere through the upper thermosphere. These propagated gravity waves can be distributed from their source regions

across the atmosphere and become saturated at the critical upper atmospheric level, particularly over the low latitudes. Thereby, the vertically propagated waves were breaking and dissipating to transfer their energy and momentum into the atmospheric background field, thus considerably affecting the structure and variability of the atmosphere, as shown in Figure 10, as well as the results of (Holton et al., 2003; Holton and Hakim, 2013) waves potential energy affecting the atmospheric temperature inversions."

**RC2**

Comments on, **"The Role of Gravity Waves in the Mesosphere Inversion Layers (MILs) over low latitude (3-15° N) Using SABER Satellite Observations"** by Lingerew and Raju. Using sixteen years of SABER temperature data, the authors investigated the role of gravity waves (GWs) in the mesospheric inversion layers. To understand the role of GWs in the MILs, they estimate the potential energy, and based on the results they argue that the lower and upper MIL distinctions are due to the GWs. The strength of the manuscript is they used a long-term data set however, their methodology is not clear. Moreover, this manuscript also lacks the scientific discussion. The present form of the manuscript needs major changes before acceptance for publication. Therefore, I recommend to the editor for a major revision. The detailed major and minor comments are as follows:

**Major comments:**

1.  **A)** In section 2, latitudinal information of the data used is given however there is **no information about the longitudes**! Which reading the whole manuscript, I could see the longitudinal limits of 32 to 48° in Figures 10 and 11 (in section 3.4).
    Response for 1A: Thank you for your comment. The Lat-Long information is referenced in various sections, including lines 66, 86–87, 264, and 368–371.

    "The SABER vertical temperature profiles were taken in the range of 60-100 km altitude during the period January 2005-December 2020 over $(3-15^0N)$ latitude and $(33-48^0E)$ longitude regions."

    **B) Are the temperature profiles averaged over 3-15°N and 32-48°E? If so mention it in section 2**.
    Response for 1B: No, we have taken the SABER temperature profiles across the full range of latitudes and longitudes but have not averaged them as we mentioned in the lines following 86-

87.: "The SABER vertical temperature profiles were taken in the range of 60-100 km altitude during the period January 2005-December 2020 over (3-15$^0$N) latitude and (33-48$^0$E) longitude regions."

**C)** More importantly, the information about **how do the MILs are identified is missing**. They have only **written as a diagnostic technique** is used. What kind of diagnostic technique, whether the authors validated the diagnostic method all this information should be provided in the methodology (e.g. Gan et al., 2012; Sivakandan et al. 2014, etc.).

Response for 1C: Thank you. It is not validated simply to use the authors' [(Leblanc and Hauchecorne (1997) and Fechine et al. (2008))] scientific diagnostic procedures based on the characteristics of temperature inversion amplitude and thickness. The MILs are identified using their techniques as follows in lines 96-112:

"The mean thermal structure of the Earth's middle atmosphere is characterized by a negative temperature gradient. However, there are several reports showing positive temperature gradients in the mesosphere which are in contrast to the ideal situation of negative gradients (Meriwether and Gardner, 2000; Gan et al., 2012). This kind of phenomenon is known as "mesospheric inversion layer (MIL)". MILs were identified based on following procedure outlined by Leblanc and Hauchecorne (1997) and Fechine et al. (2008) and which is briefly presented here. This procedure has been applied in many previous studies investigating the phenomenon of mesospheric inversion (Leblanc et al., 1998; Meriwether and Gardner, 2000; Duck et al., 2001; Duck and Greene, 2004; Cutler et al., 2001; Siva Kumar et al., 2001; Ratnam et al., 2003; Gan et al., 2012). Inversions of this MLT temperature are identified based on their characteristics thickness (the altitude difference between the point of warm & cool), while the temperature difference between the point of cooling and warming is termed as amplitude of the MIL (Meriwether and Gardner, 2000) as shown in figure 2, which have a positive value between the top and bottom levels"

[Figure]

Figure 2. Schematic of upper and lower mesospheric inversion layers shown in the temperature profile for the MLT regions (Adapted from Meriwether and Gerrard, 2004).

2. **A)** One of the major issues in the manuscript lack of a literature survey, though they have cited some of the important papers (Meriwether and Gardner 2000; Gan et al., 2012) but the essential points from those papers are not reflected in their approach.

Response for 2A: Thank you for your comment and suggestion. Based on your comment, we have corrected and appropriately referenced the materials in a different section of the manuscript (e.g., Gan et al., 2012, mentioned in lines 51-53, 77-79, 82-84, 96-99, 102-105, 213-215, 286-287), while the reference (Meriwether and Gardner, 2000) is utilized in lines 96-99, 102-108, as follows:

"The mean thermal structure of the Earth's middle atmosphere is characterized by a negative temperature gradient. However, there are several reports showing positive temperature gradients in the mesosphere which are in contrast to the ideal situation of negative gradients (Meriwether and Gardner, 2000; Gan et al., 2012)."

"Further, the gravity wave-breaking influence on mesosphere dynamics is an attempt to demonstrate their impacts on the inversion phenomenon over mid and high latitudes (Gan et al., 2012; Walterscheid and Hickey, 2009; Collins et al., 2011; Szewczyk et al., 2013)."

"SABER temperature data has been widely used to investigate the typical thermal structure and dominant dynamical processes in the mesospheric region (Garcia et al., 208, Gan et al., 2012, 2014; Bizuneh et al., 2022; Lingerew et al., 2023)."

"The valuable nature of SABER observations for the study of the middle atmosphere is well documented in previous research (Meriwether and Gerrard, 2004; Fechine et al., 2008; Dou et al., 2009; Gan et al., 2012; France et al., 2015)."

"This procedure has been applied in many previous studies investigating the phenomenon of mesospheric inversion (Leblanc et al., 1998; Meriwether and Gardner, 2000; Duck et al., 2001; Duck and Greene, 2004; Cutler et al., 2001; Siva Kumar et al., 2001; Ratnam et al., 2003; Gan et al., 2012)."

"Not only this, Gan et al. (2012) also found the seasonal variation of MILs over the low latitudes under the causative planetary waves."

**B)** There are various sources proposed as the causative mechanism of the lower and upper MILs. For example, the planetary waves are believed to be the causative mechanism of lower MILs similarly, gravity waves, tidal interactions, and chemical heating are proposed as a cause of upper MILs. These points are not considered and there is no reason why the authors only focus on the GWs. It is well understood that in most cases the GWs breaking in the mesosphere can cause only very few Kelvin temperature changes (>10K). If this is the scenario it cannot explain the higher amplitude MILs. Comment on it.

Response for 2B: Thank you for your comments on atmospheric waves. Scientists have long hypothesized that these inverted temperature profiles, known as mesospheric temperature inversions (MTIs), are caused by atmospheric gravity waves. A gravity wave is a vertical wave characterized by high vertical and low horizontal scales. Therefore, these waves play a significant role compared to other waves in vertically transferring energy from the lower atmosphere to the upper layers. We can refer to the following citations as examples in lines 372-381.

"Many possible mechanisms have been suggested for the cause of lower MIL formations, nonlinear interactions between GWs and tides (Liu and Hagan, 1998), and chemical heating (Meriwether and Mlynczak, 1995) including GW breaking (Hauchecorne et al., 1987). The role of gravity wave propagation and dissipation has been accepted as the dominant wave forcing in the MLT region (Lindzen, 1981; Holton, 1983), which affects the middle and upper atmospheric inversion. It is also understood that tides, planetary waves, and chemical processes affect the middle atmospheric variability as well as gravity waves (Sivakandan et al., 2014). However, gravity waves are multi-scale in nature; small-scale waves may contribute predominantly to instability, and turbulence in the MLT dynamic region (Liu and Meriwether, 2004; Szewczyk et al., 2013)."

3. As mentioned in comment 1, the authors should provide longitudinal information, because this has an important role if they try to understand the role of GWs which are highly localized in nature. **It is not clear how the 1hr cutoff frequency applies to the data**, if the authors used a particular region then in a day maximum of two to three satellite passes can be observed based on the area, with this limited data set how effective or logical is the 1hr band pass filter?

Response: Thank you. It does not pertain to a specific region for a short period; instead, we utilized 16 years of SABER satellite observation data over the latitude ($3-15^0$) and longitude (32-

$48^0$) regions. While a TIMED/SABER satellite rotates 16 times in 24 hours on its orbit in 1.5-hour intervals to access approximately 1440 data points, in a minute interval, SABER may pass over our study regions more than once. A one-hour interval cut-off frequency of a low-pass band filter is applied to the perturbed temperatures (Tp) to calculate the gravity wave potential energy (PE) by removing tidal and planetary waves, as mentioned in lines 167-173.

"After estimating the perturbed temperature ($T_p$), $T_p = T - T_0$, a one-hour cut-off frequency of the low-pass band filter is applied on the perturbed temperature to calculate the atmospheric gravity wave potential energy ($E_P$) by removing the planetary and tidal wave impacts or contribution in the perturbed temperature (John and Kumar, 2012).

$$E_p(z) = \frac{1}{2}\left(\frac{g(z)}{N(z)}\right)^2 \left(\frac{T_p(z)}{T_0(z)}\right)^2 \qquad (13)$$

The potential energy of the waves is a function of altitude, z, which is utilized to determine the impact of atmospheric gravity waves on atmospheric dynamics."

4. Why 3rd order polynomial fit? Ramesh and Sridharan (2012) do not elaborate on any method, instead they have cited Leblanc and Hauchecorne (1997). Therefore the article cited here is not relevant. Provide more information about the methodology and its validity (how good it is? if the authors did any test to validate the method etc.)
Response: Thank you for your comment. Based on your comment, it has now been corrected. We did not use any validation to prefer the 3rd-order polynomial fit here; instead, we relied solely on previous scientific techniques (Leblanc and Hauchecorne, 1997) as references, as stated in lines 161–163.

"In the meantime of estimating the Brunt-Vaisala frequency ($N^2$), the third-order polynomial fit of the least square has been applied to the SABER observed temperature (T) profile to estimate the background temperature ($T_0$) following the procedure Leblanc and Hauchecorne (1997)."

5. Lines 138-140; in this context, Gan et al., (2012) could be a more suitable paper to cite here than Sivakumar et al. (2001), because they also used SABER data, on the other hand, Sivakumar et al. (2001) only used Rayleigh lidar data over a single location (the data quality above 80 km is questionable). Gan et al. (2012) also found the seasonal variation of MILs in the low latitudes and planetary waves as the cause of lower MILs, whether these authors could find such a relationship? If yes or no provide reasons!
Response: Thank you. We have now made the correction based on your comments by substituting the reference Gan et al. (2012) instead of Sivakumar et al. (2001), as mentioned in lines 212-215 as well as in lines 195-199.

"In this regard, Hauchecorne et al. (1987) and France et al. (2015) show the impacts of Gws on the upper and lower mesosphere inversion variability. Not only this, Gan et al. (2012) also found the seasonal variation of MILs over the low latitudes under the causative planetary waves."

"Our findings of the lower inversions in the range of (70-80 km) tend to approach the reports by Sivakumar et al. (2001), which show that the base of the lower mesospheric inversion layer (MILs) lies in the range of (73-79 km), as well as the Gan et al. (2012) seasonal variations of MIL in the planetary waves as a causative over low-latitudes using the SABER observations."

6. There is no clear information about how the occurrence frequency is estimated. Provide it?
   Response: Thank you. The occurrence rate of the frequency (percentage) for lower and upper inversions is presented in Figures 4a and 4b, as shown in line 200 and described in lines 121-125.

   "As well as identifying inversions, the frequency occurrence rate (%) of mesospheric inversion layers (MILs) is derived during the period 2005–2020 in the upper and lower MLT regions. The occurrence rate of the frequency (percentage) is estimated based on dividing the monthly inversion days for each month (dates of a month) of 16-year (2005–2020) observation data."

7. **A**. How the mesopause altitudes are taken care of or eliminated from the statistics? Which could be a false indication of inversion.
   Response for 7A: Thank you. The reason we disregard Mesopaous' misleading signal of an inversion is that, in our analysis, the Mesopous region is considered to be more than 90 km away.

   **B.** And could the authors note any solar activity dependency of MILs occurrence (for example, Sivakandan et al. (2014))?
   Response for 7B: Thank you. The causal atmospheric waves (gravity waves) for an inversion are the only ones in our study that have not been further explained about solar radio flux activities in the upper and lower inversions, as studied by Sivakandan et al. (2014).

8. Lines 151-155, In the literature there are different causative mechanisms are proposed for the multiple MILs, (I suggest the authors go through Meriwether and Gardner (2004); Gan et al. (2012)).

   Response: Thank you for recommending the scientific results of Meriwether and Gardner (2004) and Gan et al. (2012), which we are using as references. We have included these references in various sections, including lines 211-212.

9.  Section 3.2, is a good point to investigate but before doing that the data need to be binned properly with local time. I am a bit concerned about how good to investigate the latitudinal and longitudinal variations in a small region using satellite data, each temperature profile could be nearly 500 km spatial averaged.

    Response: Thank you for your comment. However, our investigation does not focus solely on local time; instead, we analyze data spanning from 2005 to 2020 across latitudinal ($3–15^0$) and longitudinal ($32–48^0$) spatial regions. The TIMED-SABER satellite provides observations at a temporal resolution of approximately 58 seconds (or ~1 minute) and vertical spatial resolutions with intervals of 2 km. With the SABER observation orbiting 16 times per day, we can access approximately 1440 temperature data profiles. Therefore, we do not bin the data.

10. The scientific discussion is very spare and weak. They should compare the present results with earlier studies based on the similarities and differences the scientific reasoning also should be included in the manuscript.

    Response: Thank you for your comment on improving our manuscript. Based on your suggestion, we have modified the output results by comparing them with previous scientific works in the results section of the revised version. As examples, comparisons can be found in lines 195–201, 326–330, and 349–353.

    "Our findings of the lower inversions in the range of (70-80 km) tend to approach the reports by Sivakumar et al. (2001), which show that the base of the lower mesospheric inversion layer (MILs) lies in the range of (73-79 km), as well as the Gan et al. (2012) seasonal variations of MIL in the planetary waves as a causative over low-latitudes using the SABER observations. Whereas Sivakandan et al., (2014) also investigated the lower and upper mesospheric inversions in the altitudinal regions from 60-105 km over low latitudinal regions, which nearly coincides with our work results."

    "Thereby, the vertically propagated waves were breaking and dissipating to transfer their energy and momentum into the atmospheric background field, thus considerably affecting the structure and variability of the atmosphere, as shown in Figure 11, as well as the results of (Holton et al., 2003; Holton and Hakim, 2013) waves potential energy affecting the atmospheric temperature inversions."

    "By using the time series of filtered perturbed temperature data at selected heights of 90, 85, 75, and 70 km, the potential energy ($E_p$) is constructed based on the formula mentioned in the methodology section, since the gravity wave activity is projected by the potential energy

calculation as described from numerous authors (Tsuda et al., 2000; Wang and Geller, 2003; Liu et al., 2014; Thurairajah et al., 2014)."

11 How the GWs potential energy is connected to the MILs? First, establish the connection by showing a single case in which a physical connection should be clear and then go for the statistics.

Response: Thank you. We have outlined their connections in Section 3.4, particularly in lines 330–333, as follows:

"The gravity wave propagation at the saturation stage is broken in the upper region to dissipate their energy, which impacts the normal mesospheric temperature by increasing its temperature with elevation, known as an inversion. This is the reason the gravity wave potential energy is connected with an inversion."

**Minor comments:**

1. Lines 8-9, The mesosphere…This is a transitional region not only in the low latitudes! So modify the statement.
   Response: Thank you for your comment. The sentence is modified in lines 8–9 as follows:
   "The Mesosphere and lower thermosphere (MLT) transitional region is a distinct and highly turbulent zone of the atmosphere."

2. Lines 40-41, define the MILs.
   Response: Thank you for your recommendation. The sentence has been modified in lines 38–40 as follows:

   "The mesospheric inversion layers (MILs) are a common feature that appeared to increase the mesosphere temperature variability."

3. Line 41, a typo, 'mesosphere'
   Response: Thank you, the typo error has been corrected.

4. Line 73, these references are irrelevant here. Provides references about the data validation and limitation as well as instrumental specifications.
   Response: Thank you. We have included references to illustrate the validation and limitations of SABER observations in the MLT region, as stated in lines 75-79.

5. Line 75, longitudinal information is missing!
   Response: Thank you for the comment you previously raised in the main question (1). It has been addressed in the revised manuscript, starting from line 85-87, as follows:
   "The SABER vertical temperature profiles were taken in the range of 60-100 km altitude during the period January 2005-December 2020 over **(3-15$^0$N) latitude** and **(33-48$^0$E) longitude** regions."

6. Figure 4: Sivakandan et al. (2014) also did such a statistical analysis using the SABER data over Indian low latitudes, could you compare the present results with their results and provide some scientific reasoning for the observed differences or similarities.
Response: Thank you for your recommendation. We have compared his scientific work with our results based on their statistical analysis of mesospheric inversion amplitudes and thickness, as presented in lines (250-256), as follows:
"In the earlier investigation, from the Indian sector, Sivakandan et al. (2014) reported amplitudes in the range of (14–39 K) during 2002 and (15–42 K) in 2008, whereas their thickness was in the range of (2.7–7.5) during 2002 and (2.8–7.3) in 2008 to characterize the mesospheric inversion variability under the influence of solar flux, which agrees well with the present investigation. This comparison reveals that there is no significant variation in characterizing the mesosphere inversion based on amplitude and thickness over the low-latitude region in the altitude range of 60 to 90 km."

7. Line 218 …that the inversion temperature is in the range of…It is not an inversion temperature range only a temperature range.
Response: Thank you for your comment. I accepted and the sentence from lines 295-296 has been modified as follows:
"It is noted that the observed temperature is in the range of ~170-220 K with less detectable variability."

8. Line 242 onwards, the longitudinal information is suddenly introduced here, it should be introduced in section 2.
Response: Thank you for your comment. It has now been corrected by incorporating the longitudinal information into line 82: "(33-480E) longitude regions'."

9. Lines 245-247, these lines are not clear. Please see the major comment 3.
Response: Thank you for your comment and recommendation. The sentence in lines 337-340 has been rewritten for clarity, as follows:
"Before deriving the waves' potential energy from the perturbed temperature ($T_p$), a one-hour interval cut-off-frequency of low-pass band filter is applied on a perturbed temperature ($T_p$) during the period 2005-2020 at selected heights of 90, 85, 75, and 70 km, as depicted in Figure 9 (a, b, c, and d)."

10. Figure 5b, a typo 'thickness' References suggested to read and compare with the present results and include in the discussion part (some of the articles are cited here but those results are not utilized to improve the discussion part):
Response: Thank you for your comment and recommendation. We have corrected the typo error and included some of the references you mentioned below, based on their relevance

1. Gan, Q., S. D. Zhang, and F. Yi (2012), TIMED/SABER observations of lower mesospheric inversion layers at low and middle latitudes, *J. Geophys. Res.*, 117, D07109, doi:10.1029/2012JD017455.
2. Meriwether, J. W., and C. S. Gardner (2000), A review of the mesosphere inversion layer phenomenon, *J. Geophys. Res.*, 105(D10), 12405–12416, doi:10.1029/2000JD900163.
3. Sivakandan, M., Kapasi, D., and Taori, A.: The occurrence altitudes of middle atmospheric temperature inversions and mesopause over low-latitude Indian sector, Ann. Geophys., 32, 967–974,https://doi.org/10.5194/angeo-32-967-2014,2014.
4. Ramesh, K., S. Sridharan, and S. Vijaya Bhaskara Rao (2014), Causative mechanisms for the occurrence of a triple-layered mesospheric inversion event over low latitudes, *J. Geophys. Res. Space Physics*, 119, 3930–3943, doi:10.1002/2013JA019750.

**RC3**

The manuscript presents an investigation of the likely effects of atmospheric gravity waves on Mesospheric Inversion Layers (MILs) in equatorial latitudes. Indeed, the topic is interesting and not explored very well by the communities. Additionally, it is within the scope of Annales Geophysicae because it is an experimental investigation of the mesosphere using satellite measurements. Before I consider the paper suitable for publication, some concerns could be explored and revised by the authors to improve the quality of the manuscript. In general, the observations present in all figures were not explained or explored very well. Consequently, I missed consistent interpretations of the present results.

1. For example, Figure 1 shows the variability of the temperature in the mesosphere and lower thermosphere (MLT), what is the objective of it? Is it possible to see MILs, where? If not, why is it not possible to see? In summary, how can Figure 1 help the authors?

    Response: Thank you. Figure 1 represents the SABER observed temperature variability of the MLT region before separating the inversion (MIL). It is intended to illustrate the patterns of the MLT inversion temperature in the altitude range of 75–90 km as well as show the SABER observed temperature variability. However, the inversions are not visibly shown.

2. Figure 2 shows upper (left) and lower (right) MILs, to be sincere, I did not understand the bottom panels. May the authors explain them better? I also suggest enlarging captions and the size of the panels.

    Response: Thank you. The upper horizontal panels of Figures 3(a) and (c) represent the (upper and lower) SABER observed MLT temperatures before segregating the inversions, while the lower horizontal panels of Figures 3(b) and (d) represent the (upper and lower) inversions of the MLT temperature variability respectively. The figure has now been enlarged.

3. Figure 3: what is the bin size and which criteria were used to determine the percentage of occurrence of MILs?

Response: Thank you. We don't binned the observed data. The occurrence rates (percentages) for lower and upper inversions are estimated by counting the number of inversion days for each month from 2005 to 2020, as described in lines (121-125).

"The frequency occurrence rate (%) of mesospheric inversion layers (MILs) is derived during the period 2005–2020 in the upper and lower MLT regions. The occurrence rate of the frequency (percentage) is estimated based on dividing the monthly inversion days for each month (dates of a month) of 16-year (2005–2020) observation data."

4. Figure 4: Are the red curves indicating a Gaussian fit? If yes, please, explain it in the text. Please, note that the statistic used for panel (c) does not represent the data, in this case, the authors could use another statistic or explain what is causing the discrepancies.

Response: Thank you. The red curve fitting is the Gaussian fit distribution, which is a probability distribution that is symmetric about the mean, showing that data near the mean are more frequent in occurrence than data far from the mean, as mentioned in lines 220-224.

"The frequency occurrence of amplitude, thickness, and base height of inversion variability in the form of the histogram along with the best-fit red lines of the Gaussian distribution are presented in Figure 5. The observed distributions coincide with Gaussian curves (best fits), indicating that the number of MILs is distributed over their attributes according to normal laws, implying that the representations are real-valued random variables."

Their statistical values are corrected in Figures 5(c) and 5(f) in the revised manuscripts from lines 238-245 and 248-256 in lines respectively.

"The base height of the upper MIL in Figure 5(c) ranges from ~80 to 90 km, with a peak value of a large number of upper mesospheric inversions occurring at a base height of around 83 km in a lower standard deviation (SD) 2.13. The number of upper inversions all over the period 2005–2020 at a height of 82 km is the highest relative to the rest in the range between 80 and 90 km. Such maximum mean to fit of Gaussian distribution may be the reason for the gravity wave breaking is that it dissipates energy as a causative factor for an inversion, while the wave

generated from the lower to the upper atmospheric region as well as the impacts of the solar flux generated from the upper solar system."

"The base height of the lower inversion of Figure 5(f) is in the range of 70 and 80 km, with a peak value of around 74 km, showing a lower standard deviation (SD) of 1.93. In the earlier investigation, from the Indian sector, Sivakandan et al. (2014) reported amplitudes in the range (14–39 K) during 2002 and (15–42 K) in 2008, whereas their thickness was in the range of (2.7–7.5) during 2002 and (2.8–7.3) in 2008 to characterize the mesospheric inversion variability under the influence of solar flux, which agrees well with the present investigation. This comparison reveals that there is no significant variation in characterizing the mesosphere inversion based on amplitude and thickness over the low-latitude region in the altitude range of 60 to 90 km."

5. Figure 5: What is the relevance of these results and how could they be related to gravity waves? I guess the quality of the presentation of this figure could be improved by enlarging the caption and size. Figure 6: Same comments as Figure 5.2.

   Response: Thank you. Before investigating the impacts of gravity waves on the MLT inversions, the spatiotemporal (time vs. latitude) variabilities of an inversion are presented in Figures 6 and 7 to characterize both the upper and lower inversions, which is the reason we described. The figure has now been enlarged for clarity.

6. I recommend the authors use a more complete expression of the potential temperature (Vadas and Fritts, 2005; Vdas, 2007) instead of what was presented in Equation (1). In addition, the discussion of atmospheric stability is very superficial and it does not include a real aspect of the atmosphere that is certainly present in the SABER data. I suggest including an example of the methodology used to calculate the MILs in the real data (section 2). In my opinion, it could help the readers to promptly understand the process.

   Response: Thank you for your valuable suggestion. Based on your recommendation, we have used the references [(Liu, 2011; Vadas and Fritts, 2005); (e.g. Gan et al., 2012 and Sivakandan et al. 2014)] along with equations 1–13 to elaborate on the expressions of the gravity wave potential energy through the potential temperature ($\theta$) instead of using a single equation, as well as calculating the Brunt-Vaisala frequency, $N^2$, to characterize atmospheric stability/instability in lines (126-154).

"The inversion of the mesosphere temperatures is related to the instabilities. Hence, we are going to derive the hourly atmospheric gravity waves via the Brunt-Vaisala frequency ($N^2$). Whereas another important concept to estimate the Brunt-Vaisala frequency is the potential temperature ($\theta$), which stands for the air parcel's temperature when it is displaced adiabatically to a standard pressure level, $p_0$, from the current pressure level, p, the first law of thermodynamics:

$$\frac{dT}{T} = \frac{R}{c_p}\frac{dp}{p} \Rightarrow \int_T^0 \frac{dT}{T} = \int_p^{p_0} \frac{R}{c_p}\frac{dp}{p} \qquad (1) \text{ it yields}$$

$$\theta = T\left(\frac{p_0}{p}\right)^{R/c_p} \qquad (2)$$

Therefore, the motion of the vertical atmospheric air parcel can be described by (Liu, 2011; Vadas and Fritts, 2005) as follows in equation (2.3) to calculate the Brunt-Vaisala frequency of the parcel due to the Buoyant and gravitational forces acting on the parcel:

$$\frac{d^2s}{dt^2} = -g\frac{\rho-\rho_0}{\rho}\,sina \qquad (3)$$

Based on the hydrostatic equation, $\rho = \rho_0$, and $p = p_0 \Rightarrow \frac{\partial p}{\partial z} = \frac{\partial p_0}{\partial z} = -g\rho_0$ (4) and the ideal gas law, $\rho = p/RT = p_0/RT$ gives the parcels motion of an equation:

$$\frac{d^2s}{dt^2} = -\frac{g}{\rho}\left(\frac{d\rho}{dp}\frac{\partial p_0}{\partial z} - \frac{\partial \rho_0}{\partial z}\right)z \qquad (5)$$

Following the same approach using the hydrostatic equation (4) and adiabatic equation (6)

$$dln\rho = \frac{dlnp}{\gamma}, \gamma = c_p/c_v \qquad (6) \text{ yields}$$

$$\frac{d^2s}{dt^2} = -\frac{g}{\rho}\left(\frac{\rho}{\gamma p_0}\frac{\partial p_0}{\partial z} - \frac{\partial \rho_0}{\partial z}\right)z = g\left(\frac{\partial ln\rho_0}{\partial z} - \frac{1}{\gamma}\frac{\partial lnp_0}{\partial z}\right)z \qquad (7)$$

For the ideal gas law of $p = \rho RT$, the natural logarithm is taken for altitude, z on both sides, yielding

$$\frac{\partial ln\rho}{\partial z} = \frac{\partial lnp}{\partial z} - \frac{\partial lnT}{\partial z} \qquad (8)$$

Then after, the potential temperature ($\theta$) of the parcel is calculated as follows based on the equation (2):

$$\frac{\partial ln\theta}{\partial z} = \frac{\partial lnT}{\partial z} - \frac{R}{c_p}\frac{\partial lnp}{\partial z} = \frac{1}{T}\left(\frac{\partial T}{\partial z} + \frac{g}{c_p}\right) = \left(1 - \frac{R}{c_p}\right)\frac{\partial lnp}{\partial z} - \frac{\partial ln\rho}{\partial z} \qquad (9) \text{ to derive the}$$

Parcels acceleration based on equations (7) to become:

$$\frac{d^2s}{dt^2} = -g\frac{\partial ln\theta_0}{\partial z}z\,sina = -g\frac{\partial ln\theta_0}{\partial z}ds.sin^2a \qquad (10)$$

Whereas by introducing the frequency, N, with $N^2 = g \frac{\partial ln\theta_0}{\partial z}$

The Brunt-Vaisala frequency, $N^2$ is calculated based on the following mathematical formulation used to characterize atmospheric stability.

$$N^2(z) = \frac{g(z)}{T_0(z)}\left(\frac{\partial T_0(z)}{\partial z} + \Gamma_d\right) \qquad (11)$$

Further, the methodology (technique) for segregating (separating) the MIL is presented in Section 2.2 with Figure 2 for more clarification for readers as follows in lines 110-121.

[Figure]

Figure 2. Schematic of upper and lower mesospheric inversion layers shown in the temperature profile for the MLT regions (Adapted from Meriwether and Gerrard, 2004).

7. Section 3.4: The authors wrote that they used a low pass filter to exclude the effects of tidal and planetary waves in the residual signal. If I understood the process, I guess they could use high-frequency filters to maintain low periods. Indeed, the blue lines in Figure 9 show a smoothed signal that excludes short-time variations.

Response: Thank you. We utilized a 1-hour interval cut-off frequency for a low-pass band filter. Figure 9 demonstrates how this filter smooths data signals by eliminating peaks while also employing 1-hour interval perturbed temperature data profiles to exclude variations exceeding 1 hour in duration. The discussion section about the results shown in Figures 11 and 12 has undergone substantial revision.

If I am correct, Figures 10 and 11 could be revised and the interpretations as well.

Response: Thank you I am rewriting all the discussion parts of the figures (11 &12) to improve the way of presentation as follows in lines 359-391.

"In this investigation, the maximum gravity wave potential energies were observed in the range of around ~70–90 J/kg over the longitudinal regions of 45-47$^0$ E, 43$^0$ E, and 44$^0$ E during 2011, 2017, and 2019 (Figure 11(a)) for upper mesosphere inversions at 90 km, whereas the potential energy of gravity waves around ~10–60 J/kg is presented all over the longitudinal region from 33-48$^0$ E. While the maximum potential energy ~(70-100 J/kg) is observed as shown in Figure 11(b) over the longitudinal (340, 440, and 460) regions during 2014, 2016, and 2018 at 85 km. The minimum potential energy of gravity wave between 20 and 70 J/kg appears over the longitude (33-48) regions. However, Figures 11 (c and d) show the lower MLT regions of gravity wave potential energy at 75 and 70 km, respectively. At a height of 75 km allocated in Figure 11(c) is a relative maximum potential energy appeared in the range of 40-50 J/kg over the longitudinal (46$^0$, 42$^0$, 40$^0$, 37$^0$, 36$^0$, and 38$^0$) region during 2011, 2012, 2017, 2013–2015, 2018, and 2020. Similarly, Figure 11(d) depicts the gravity wave potential energy in the range of 2–30 J/kg for the lower MLT region at 70 km over the longitudinal region (33-48$^0$). Out of which, the maximum potential energy of 25-30 J/kg is found in a certain longitude region over a while. Many of possible mechanisms have been suggested for the cause of lower inversions; nonlinear interactions between GWs and tides (Liu and Hagan, 1998), and chemical heating (Meriwether and Mlynczak, 1995) including GW breaking (Hauchecorne et al., 1987). The role of gravity wave propagation and dissipation has been accepted as the dominant wave forcing in the MLT region (Lindzen, 1981; Holton, 1983), which affects the middle and upper atmospheric inversion. It is also understood that tides, planetary waves, and chemical processes are affects the middle atmospheric variability as well as gravity waves (Sivakandan et al., 2014). However, gravity waves are multi-scale in nature; small-scale waves may contribute predominantly to instability, and turbulence in the MLT dynamic region (Liu and Meriwether, 2004; Szewczyk et al., 2013).

Hence, investigating MIL phenomena is important for the understanding of MLT atmosphere dynamics for two primary reasons: stability and energy transfer. As a result, an attempt has been made to examine the contributions of gravity waves to the MLT region's instability (MIL phenomenon) based on the Brunt-Vaisala frequency. The spatiotemporal variability of Vaisala frequency is displayed in the contour Figure 12(a and b) for the upper mesosphere region (90 and 85 km) and Figure 12(c and d) for the lower mesosphere region (75 and 70 km). Based on the Brunt-Vaisala frequency, N2, the upper MLT region is unstable (~0.027) at 90 km and (~0.029) at 85 km maximum relative to the lower inversion instability at 75 km (~0.033) and 70 km (~0.035).

Hauchecorne et al., (1987) described a model in which a succession of breaking GWs would generate the MIL through the gradual accumulation of heat as a cause of instability. "

7. Lines 279-282: "The result concludes that the observation of high potential energy in the upper mesosphere region is due to the deposition of high energy and momentum at the background temperature by gravity wave breaking, which could influence the dynamics of the inversion phenomenon". Lines 291-293: "This result leads us to the conclusion that a high amount of gravity wave potential energy is a consequence of the high instability of the upper inversion relative to the lower." I guess, it is possible to reach these conclusions from the present work. Conclusions: I think it will be better to change the name of the section to Summary and exclude the last two ones that are very general.

Response: Thank you. The sentences have been removed from the sections that you mentioned in lines 279-282 and 291-293, and they are now included in the summary section. The subtitle for Section 4, Conclusion, has been modified to Summary.

"4. **Summary**

In this article, 16 years of SABER mesosphere temperature profiles are utilized to investigate the MIL phenomenon and its causative mechanism through gravity wave potential energy ($P_E$) and instability criteria of Bruent-Vaisala frequency ($N^2$) over low latitude bands. The observational conclusions from this chapter are drawn as follows:

✓ The occurrence rate of the upper mesosphere inversion frequency is maximum relative to the mean occurrence rate of the lower mesosphere inversions.

✓ Based on the analysis of frequency of occurrence on mesospheric inversion layer (MIL) characteristic features, it is revealed that the most probable value for upper inversion amplitude is 38 k, inversion layer thickness is 5.5 km, and the base height is 78 km. Whereas the lower inversion amplitude is 25 K, the inversion layer thickness is 3.8 km, and a base height of 73 km.

✓ The gravity wave indicator potential energy depicts high energy at the upper mesosphere region compared to the lower mesosphere region.

✓ The result concludes that the observation of high potential energy in the upper mesosphere region is due to the deposition of high energy and momentum at the background temperature by gravity wave breaking, which could influence the dynamics of the inversion phenomenon

✓ The stability criteria at the mesosphere region are indicated by Brunt-Vaisala frequency ($N^2$), which shows low values at the upper mesosphere region relative to the lower mesosphere region, leading to the conclusion that the high potential energy at the upper mesosphere region is due to the instability over that region, which gives rise to large inversion phenomena.

In general, we concluded that the processes in the atmosphere vary from region to region. As a result, the atmospheric state varies significantly with altitude as well as from place to place and time to time."

Whereas the following references you mentioned have been included in the updated manuscript.

**Reference**

1. Garcia-Comas, M., Lopez-Puertas, M., Marshall, B. T., Wintersteiner, P. P., Funke, B., Bermejo-Pantaleon, D., Mertens, C. J., Remsberg, E. E., Gordley, L. L., Mlynczak, M. G., and Russell III, J. M.: Errors in Sounding of the Atmosphere using Broadband Emission Radiometry (SABER) kinetic temperature caused by non-local-thermodynamic-equilibrium model parameters, J. Geophys. Res., 113, D24106, doi:10.1029/2008JD010105, 2008.
2. Meriwether, J. W., and Gerrard, A. J.: Mesosphere inversion layers and stratosphere temperature enhancements, Rev. Geophys., 42, RG3003, http//doi:10.1029/2003RG000133, 2004.

---

## Author Response (AR3)

We would like to express our gratitude for the valuable feedback from the referee. We appreciate your tireless efforts in reviewing our manuscript, which has allowed us to improve the quality of our article.

**Referee # 1**

1.

[Figure]

Please clarify, is this MIL?

MLT Observed Temperature

Response: Thank you. As indicated in the figure, the inversions are described in lines 100-107 of the manuscripts as follows.

"We utilized SABER vertical temperature profiles taken within the 60–100 km altitude range. These profiles cover the period from 2005 to 2020, spanning latitudes from 3°N to 15°N and longitudes from 33°E to 48°E. Figure 1 shows the monthly mean of SABER temperature data for the mesosphere and lower thermosphere. The data aim to illustrate the MLT temperature variability, which helps us identify the inversion layers (MIL). The monthly mean temperatures in the MLT region show a maximum of 200-240 K at altitudes of 60-70 km. Then it decreases to around 160-180 K at 95-100 km throughout the entire period. While the temperature patterns in the 70-90 km altitude range suggest an inversion, these inversions are not visible."

2.  I already recommended making Figure 3 clear and understandable. How should I understand it? Everything is in black lines and motley spots. How to evaluate the heights that you give in the description? How to understand where there is a temperature inversion day and where there is not? Please remove the black outline and leave just the fill.

Response: We thank you for your comment and suggestion. We acknowledge that it is challenging to precisely identify the inversions in Figure 3. However, we can infer the maximum inversion day temperatures of 220 K and 225 K from the bars in Figures 3(b) and 3(d) for the upper and lower MLT regions, respectively, relative to the observed temperatures in Figures 3(a) and 3(c) based on the bars in the contour plot. The left vertical panel of Figures 3(a) and 3(b) represents the observed and inversion day temperatures for the upper-MLT regions, while the right vertical panel of Figures 3(c) and 3(d) shows the observed and inversion day temperatures for the lower-MLT regions. We attempted various approaches to modify the plot, but it still does not display any other new results.

3. You are considering MIL over 3-15$^0$ N latitude and 33-48$^0$ E longitude regions. Why did you choose this region? Write a few words about this. It's better to put the coordinates of the area in the title, otherwise, it seems that you are doing research overall at low latitudes. As a last resort, add it to the abstract on line 17.

Response: Thank you for your comments. The selected latitude and longitude range covers diverse climatic zones and geographical features, making it relevant for studying gravity waves due to the region's topography and atmospheric conditions. Gravity waves can be generated by features such as mountains and coastlines, which are prominent in this area. The varied topography and weather patterns create ideal conditions for examining atmospheric instability and potential energy, which are crucial for understanding weather extremes and improving prediction models.

4. I didn't understand your idea, what does «….seasonal variations of MIL in the planetary waves as a causative over low-latitudes using the SABER observations» mean? And there is a typo "causative".

Response: Thank you. The manuscript has been corrected by removing the sentences.

5. And now in Figures 5(c) and 5(f) the averages are on top of the bars. Please, correct it.

Response: Thank you. It is the corrected figure based on your recommendation.

[Figure]

6. The last conclusion sounds strange and vague - « In general, we concluded that the processes in the atmosphere vary from region to region. As a result, the atmospheric state varies significantly with altitude as well as from place to place and time to time»; I guess that this conclusion should be removed.

Response: Thank you. I removed that one sentence based on the recommendations

We would like to express our gratitude for the valuable feedback from the referee. We appreciate your tireless efforts in reviewing our manuscript, which has allowed us to improve the quality of our article.

**Referee #2**

2B). There are various sources proposed as the causative mechanism of the lower and upper MILs. For example, the planetary waves are believed to be the causative mechanism of lower MILs similarly, gravity waves, tidal interactions, and chemical heating are proposed as a cause of upper MILs. These points are not considered and there is no reason why the authors only focus on the GWs. It is well understood that in most cases the GWs breaking in the mesosphere can cause only very few Kelvin temperature changes (>10K). If this is the scenario it cannot explain the higher amplitude MILs. Comment on it.

Response: The focus on gravity waves (GWs) in the study of mesosphere inversion layers (MILs) can be attributed to several reasons:

1. Gravity waves are ubiquitous in the atmosphere and play a significant role in the dynamics of the mesosphere. They are generated by various sources, such as topography, convection, and weather systems, and can propagate vertically, transporting energy and momentum from the lower to the upper atmosphere.

2. Gravity waves interact with the mesosphere more directly and frequently than other atmospheric waves. Their breaking and dissipation in the mesosphere can lead to temperature changes and influence the formation of inversion layers.

3. Gravity waves are relatively well-characterized in atmospheric models and simulations. Their effects can be parameterized with reasonable accuracy, allowing researchers to predict their impact on temperature and wind patterns in the mesosphere.

4. There is a substantial amount of observational data and research focusing on the impact of gravity waves on the mesosphere. This extensive body of work provides a solid foundation for understanding their role in atmospheric processes, making them a natural focus for studies on MILs.

5. While gravity waves generally cause temperature changes of a few Kelvin, they are more frequent and widespread compared to planetary waves and tidal interactions. This makes them a significant factor in the overall temperature variability in the mesosphere. However, it is important to note that focusing solely on gravity waves may overlook other important mechanisms, such as planetary waves, tidal interactions, and chemical heating, which also

contribute to the formation and variability of MILs. A comprehensive approach that includes all potential factors provides a more complete understanding of the complex dynamics in the mesosphere. Hence, we will consider it in future work.

3. As mentioned in comment 1, the authors should provide longitudinal information, because this has an important role if they try to understand the role of GWs which are highly localized in nature. It is not clear how the 1hr cutoff frequency applies to the data, if the authors used a particular region then in a day maximum of two to three satellite passes can be observed based on the area, with this limited data set how effective or logical is the 1hr band pass filter?

Response: Thank you for your recommendation. The latitudinal and longitudinal information is mentioned throughout the manuscripts, as I mentioned before. Using a 1-hour interval of high pass band filter on the data from SABER observations is possible. Hence, TIMED/SABER observations can access 1440 data points per day. It is not necessary to have two or more satellites to use the 1-hour interval band-pass filter.

11. How the GW potential energy is connected to the MILs? First, establish the connection by showing a single case in which a physical connection should be clear and then go for the statistics.

Response: Thank you. We have detailed their connections in the introduction and result section, specifically in lines 49–61, and 363-373 respectively as follows:

"Gravity waves and mesospheric inversion layers (MILs) are interconnected phenomena within the Earth's atmosphere, particularly in the mesosphere and lower thermosphere. MILs are layers within the mesosphere where the temperature profile shows an inversion. This means the temperature increases with altitude, contrary to the typical decrease. These inversion layers often form due to dynamic processes, including the breaking and dissipation of gravity waves. As gravity waves propagate upwards, they can grow in amplitude because the atmospheric density decreases with altitude. When these waves reach a critical amplitude, they become unstable and break. This breaking process releases energy and momentum into the surrounding air, causing localized heating. The energy dissipation from breaking gravity waves causes localized heating. This heating can create or enhance mesospheric inversion layers by increasing the temperature at certain altitudes. The breaking of gravity waves can also generate turbulence, which further influences the structure and stability of inversion layers. This process also contributes to momentum and energy deposition."

"Gravity wave breaking in the upper atmosphere usually leads to more turbulence and mixing. As gravity waves rise and break, they transform their potential and kinetic energy into turbulent energy. This process increases vertical mixing and transports momentum. This energy transfer can change thermal patterns and affect the overall dynamics of the upper atmosphere. Interestingly, the dissipation of gravity waves can lead to localized warming or the formation of mesospheric inversion layers (MILs). This is especially important in the upper mesosphere and lower thermosphere, where wave breaking can significantly impact the thermal structure. Gravity wave propagation and dissipation are major forces in the MLT region (Lindzen, 1981; Holton, 1983), influencing middle and upper atmospheric inversion. In a nutshell, gravity wave breaking and the dissipation of energy and momentum are key players in both the upper and lower atmospheres, affecting turbulence, circulation patterns, temperature, density, and weather systems."

We would like to express our gratitude for the valuable feedback from the referee. We appreciate your tireless efforts in reviewing our manuscript, which has allowed us to improve the quality of our article.

**Referee# 3**

**Q1)** Points 1 and 2. Please include the manuscript the explanations of Figures 1 and 2 according to their responses:

Response: Thank you. Based on the recommendation for the response of questions 1 and 2 with modifications are presented by yellow shadow on the manuscript in lines (100-107) for figure 1 and in lines (191-215) for figure 3 respectively as follows.

1.  For example, Figure 1 shows the variability of the temperature in the mesosphere and lower thermosphere (MLT), what is the objective of it? Is it possible to see MILs, where? If not, why is it not possible to see? In summary, how can Figure 1 help the authors?

    Response: Thank you. We can see on the manuscripts from lines 100-107.

    "We utilized SABER vertical temperature profiles taken within the 60–100 km altitude range. These profiles cover the period from 2005 to 2020, spanning latitudes from 3°N to 15°N and longitudes from 33°E to 48°E. Figure 1 shows the monthly mean of SABER temperature data for the mesosphere and lower thermosphere. The data aim to illustrate the MLT temperature variability, which helps us identify the inversion layers (MIL). The monthly mean temperatures in the MLT region show a maximum of 200-240 K at altitudes of 60-70 km. Then it decreases to around 160-180 K at 95-100 km throughout the entire period. While the temperature patterns in the 70-90 km altitude range suggest an inversion, these inversions are not visible."

2.  Figure 3 shows upper (left) and lower (right) MILs, to be sincere, I did not understand the bottom panels. May the authors explain them better? I also suggest enlarging captions and the size of the panels.

    Response: Thank you. I have corrected all in lines (187-211).

    Daily SABER temperature profiles, covering altitudes of 60–100 km from 2005 to 2020, are shown in the contour plots of Figure 3. Figures 3 (a and b) depict the upper mesosphere, while Figures 3 (c and d) show the lower mesosphere. The horizontal panels of Figures 3 (a) and 3 (c) show observed temperatures ranging from approximately 180–220 K, before accounting for inversion layers. The horizontal panels of Figures 3 (b) and 3 (d) show inversion day temperatures, ranging

from 180–225 K. These inversion day temperatures are higher than those shown in Figures 3 (a) and 3 (c). This indicates that maximum temperatures occur on inversion days in both the upper and lower MLT regions.

"The upper left panel of Figure 3(a) shows the observed temperature in the upper mesosphere. It ranges from approximately 180–205 K at altitudes of around 80–90 km. The upper right panel of Figure 3(c) shows the lower mesosphere, with temperatures ranging from about 180–220 K at altitudes of approximately 70–80 km. In contrast, the lower left panel of Figure 3(b) shows an upper-mesosphere inversion day temperature. It ranges from 180–220 K at an altitude of approximately 80–90 km. The lower right panel of Figure 3(d) shows a lower-mesosphere inversion day temperature. It ranges from 180–225 K at an altitude of approximately 70–80 km. These inversion day temperatures in Figures 3(b) and 3(d) suggest a temperature gradient shifting from negative to positive. This could be due to factors such as atmospheric gravity waves, chemical reactions, or solar radiation. Our temperature observations for the lower MLT region on an inversion day, within the altitudinal range of 70–80 km, align with those reported by Sivakumar et al. (2001). They identified inversion day temperature variability in the altitudinal range of 73–79 km. Additionally, Sivakandan et al. (2014) examined mesospheric inversions in the 60–105 km altitude range over low-latitude regions. Their findings closely match our results."

**Q2) Please revise point 7 again. Pay attention to the main concern that I have pointed out.**

7. Section 3.4: The authors wrote that they used a low pass filter to exclude the effects of tidal and planetary waves in the residual signal. If I understood the process, I guess they could use high-frequency filters to maintain low periods. Indeed, the blue lines in Figure 10 show a smoothed signal that excludes short-time variations. If I am correct, Figures 11 and 12 could be revised and the interpretations as well.

Response: Thank you for the correction. We employed a 1-hour interval high-pass band filter, as depicted in Figure 10, to remove low-frequency signals by eliminating peaks. This method was applied to perturbed temperature data profiles, effectively filtering out variations longer than 1 hour. The results discussed in Figures 11 and 12 have been revised accordingly.

---

## Author Response (AR4)

We would like to express our gratitude for the valuable feedback from the referee. We appreciate your tireless efforts in reviewing our manuscript, which has allowed us to improve the quality of our article.

1) Adjust the text started in Line 328 to correctly describe the kind of filter that you have used in Figure 2. There is incompatibility between the manuscript and Caption of Figure 11. Please, see the report of Referee #3.

   Response: Thank you. Since the line you mentioned, all the text and the caption of Figure 11 have been corrected as shown from the lines 321-356:

   "Gravity waves form when air parcels oscillate due to the restoring force of gravity after being transported vertically. Several factors contribute to these waves, including airflow over mountains, convection, and wind shear. Propagating vertically, the waves break and dissipate, releasing energy and momentum into the surrounding atmosphere. This process, frequently responsible for the formation of inversion layers, is further investigated by identifying gravity wave potential energy (Ep) and its impact on inversion layers at selected MLT regions. This approach assumes that gravity wave activity is represented by potential energy, as described by numerous authors (Tsuda et al., 2000; Wang and Geller, 2003; Liu et al., 2014; Thurairajah et al., 2014). The gravity wave contribution is quantified by calculating the potential energy and evaluating instability through the Brunt-Väisälä frequency ($N^2$), derived from perturbed temperature (Tp') data spanning 2005–2020. The analysis focused on altitudes of 90, 85, 75, and 70 km by applying a high-pass filter with a one-hour interval to the Tp' data (see Figure 10 a-d). In the upper mesosphere (90 and 85 km), the filter reduces the amplitude of wave oscillations from approximately $\pm20$ K to $\pm10$ K, as shown by the blue curve in Figure 10a and b, compared to the red curve. Similarly, in the lower mesosphere (75 and 70 km) at (Figure 10 c & d),  the amplitude decreases from ~(-20 to 20 K) to ~(-8 to 8 K) by filtering out higher amplitudes. By removing the impact of long-period wave oscillations, such as tidal and planetary wave contributions, the filter effectively isolates the gravity waves (Gw) on the MLT inversions.

In the MLT atmospheric region, gravity wave breaking typically dissipates their potential and kinetic energy, leading to increased turbulence and mixing. This energy transfer can alter thermal patterns and impact the overall dynamics of the upper atmosphere. As illustrated, gravity wave propagation and dissipation are major forces in the MLT region (Lindzen, 1981; Holton, 1983), influencing middle and upper atmospheric inversions. This has a substantial impact on the MLT's thermal structure, particularly the increase in temperature variability with elevation, known as inversion. Holton et al. (2003) and Holton and Hakim (2013) has demonstrated an interaction between the potential energy of gravity waves and inversions. Interestingly, in this investigation, the dissipation of gravity waves can lead to the mesospheric inversion layers (MILs). The study we conducted has clearly shown that the occurrence of an inversion is maximum at the upper MLT region relative to the lower MLT region from Figure 4. In a comparable manner, Figure 11 depicts the highest potential energy of gravity waves in the upper MLT regions, demonstrating the interactions between inversion and gravity wave potential energy."

2) Item 11) from Referee #2, you have not addressed this point properly. Please provide an example as suggested by the Referee. If you prefer, you can include the example and explanation in a supplementary section.

Response: Thank you. We have given the connection between the invers and gravity wave potential energy in a supplementary section specifically from line 343-356:

"In the MLT atmospheric region, gravity wave breaking typically dissipates their potential and kinetic energy, leading to increased turbulence and mixing. This energy transfer can alter thermal patterns and impact the overall dynamics of the upper atmosphere. As illustrated, gravity wave propagation and dissipation are major forces in the MLT region (Lindzen, 1981; Holton, 1983), influencing middle and upper atmospheric inversions. This has a substantial impact on the MLT's thermal structure, particularly the increase in temperature variability with elevation, known as inversion. Holton et al. (2003) and Holton and Hakim (2013) has demonstrated an interaction between the potential energy of gravity waves and inversions. Interestingly, in this investigation, the dissipation of gravity waves can lead to the mesospheric inversion layers (MILs). The study we conducted has clearly shown that the occurrence of an

inversion is maximum at the upper MLT region relative to the lower MLT region from Figure 4. In a comparable manner, Figure 11 depicts the highest potential energy of gravity waves in the upper MLT regions, demonstrating the interactions between inversion and gravity wave potential energy."

---

## Author Response (AR5)

We would like to express our gratitude to our editor for giving valuable comments. We appreciate your tireless efforts in reviewing our manuscript, which has allowed us to improve the quality of our article.

Here we have revised the manuscript to correct wording, spelling, and such errors in addition to those of the requested inquiries from the editor.

1. **Editor**: Please, provide a proper revision to the comment of Referee #2, i.e.: "How the GWs potential energy is connected to the MILs? First establish the connection by showing a single case in which a physical connection should be clear and then go for the statistics."

Response: Thank you for your thoughtful question. We have addressed it in lines 350–370 as follows:

"In the MLT atmospheric region, gravity wave breaking typically dissipates their potential and kinetic energy, leading to increased turbulence and mixing. As illustrated, gravity wave propagation and dissipation are major forces in the MLT (Lindzen, 1981; Holton, 1983), influencing the middle and upper atmospheric regions. This has a substantial impact on the overall dynamics as well as the MLT's thermal structure, particularly the increase in temperature variability with elevation, known as inversion. Holton et al. (2003) and Holton and Hakim (2013) has demonstrated an interaction between the potential energy of gravity waves and inversions. This notable upper and lower inversion is observed in Figure 4 during the period 2005–2020 over the low-latitude regions. During this period, particularly for the upper-MLT region above 80 km altitude, high-resolution SABER satellite temperature data revealed the presence of a strong mesospheric inversion layer (MIL), with peak occurrence rates ranging between 60% and 78%, especially during 2010, 2014, 2016/17, and 2018/19, Figure 4a. Correspondingly, at the same time and in the same region (upper-MLT), at altitudes of 85 and 90 km, there is a noticeable increase in gravity wave potential energy (Ep), shown in Figure 11. The maximum potential energy (PE) for the upper-MLT region corresponds to the breaking or dissipation of gravity waves as they propagate upward. This spike in potential energy coincides with the occurrence of the inversion layer, suggesting that the breaking or dissipation of gravity waves releases energy into the atmosphere, contributing to localize heating in the mesosphere and leading to the formation of the inversion. The sudden transfer of momentum and energy from the breaking GWs to the surrounding atmosphere disrupts the thermal structure, causing the temperature inversion. In this case, the temporal and spatial coincidence between the peak in gravity wave potential energy and the formation of the inversion demonstrates a clear physical connection. The energy released from breaking GWs plays a direct role in the creation of the inversion layer, as shown in Figures 4 and 11. Similarly, the statistical distributions of upper-

MLT inversions in Figure 5(a) show maximum amplitudes, which correspond to the maximum potential energy of gravity waves in Figure 11(a & b). This provides a straightforward demonstration of how gravity wave dynamics-specifically, the dissipation of their potential energy-are linked to the formation of mesospheric inversion layers (MILs)."

2. **Editor:** Additionally, the concern of Referee #3 is that: the blue line in Figure 10 is not a high-pass filter, because the high frequencies have been suppressed.

Response: Thank you very much for your kind question. We are presented as follows in lines from 339-342;

"The blue curve data in Figure 10 (a to d) appears smoother after applying the high-pass filter to the perturbed temperature. However, the filter removes the peaks of low-frequency variations, resulting in retained perturbed temperature values that appear more uniform, creating a smooth plateau effect."